# Towards Understanding and Mitigating Dimensional Collapse in Heterogeneous Federated Learning

**Yujun Shi**[1] *   **Jian Liang**[3]   **Wenqing Zhang**[2]   **Vincent Y. F. Tan**[1]   **Song Bai**[2]
[1]National University of Singapore   [2]ByteDance Inc.   [3]Institute of Automation, CAS
shi.yujun@u.nus.edu   vtan@nus.edu.sg   songbai.site@gmail.com

## Abstract

Federated learning aims to train models collaboratively across different clients without sharing data for privacy considerations. However, one major challenge for this learning paradigm is the *data heterogeneity* problem, which refers to the discrepancies between the local data distributions among various clients. To tackle this problem, we first study how data heterogeneity affects the representations of the globally aggregated models. Interestingly, we find that heterogeneous data results in the global model suffering from severe *dimensional collapse*, in which representations tend to reside in a lower-dimensional space instead of the ambient space. Moreover, we observe a similar phenomenon on models locally trained on each client and deduce that the dimensional collapse on the global model is inherited from local models. In addition, we theoretically analyze the gradient flow dynamics to shed light on how data heterogeneity result in dimensional collapse for local models. To remedy this problem caused by the data heterogeneity, we propose FedDecorr, a novel method that can effectively mitigate dimensional collapse in federated learning. Specifically, FedDecorr applies a regularization term during local training that encourages different dimensions of representations to be uncorrelated. FedDecorr, which is implementation-friendly and computationally-efficient, yields consistent improvements over baselines on standard benchmark datasets. Code: https://github.com/bytedance/FedDecorr.

## 1 Introduction

With the rapid development deep learning and the availability of large amounts of data, concerns regarding data privacy have been attracting increasingly more attention from industry and academia. To address this concern, McMahan et al. (2017) propose *Federated Learning*—a decentralized training paradigm enabling collaborative training across different clients without sharing data.

One major challenge in federated learning is the potential discrepancies in the distributions of local training data among clients, which is known as the *data heterogeneity* problem. In particular, this paper focuses on the heterogeneity of *label distributions* (see Fig. 1(a) for an example). Such discrepancies can result in drastic disagreements between the local optima of the clients and the desired global optimum, which may lead to severe performance degradation of the global model. Previous works attempting to tackle this challenge mainly focus on the model parameters, either during local training (Li et al., 2020; Karimireddy et al., 2020) or global aggregation (Wang et al., 2020b). However, these methods usually result in an excessive computation burden or high communication costs (Li et al., 2021a) because deep neural networks are typically heavily over-parameterized. In contrast, in this work, we focus on the representation space of the model and study the impact of data heterogeneity.

To commence, we study how heterogeneous data affects the global model in federated learning in Sec. 3.1. Specifically, we compare representations produced by global models trained under different degrees of data heterogeneity. Since the singular values of the covariance matrix provide a comprehensive characterization of the distribution of high-dimensional embeddings, we use it to

---

*Work done when interning with Song Bai.

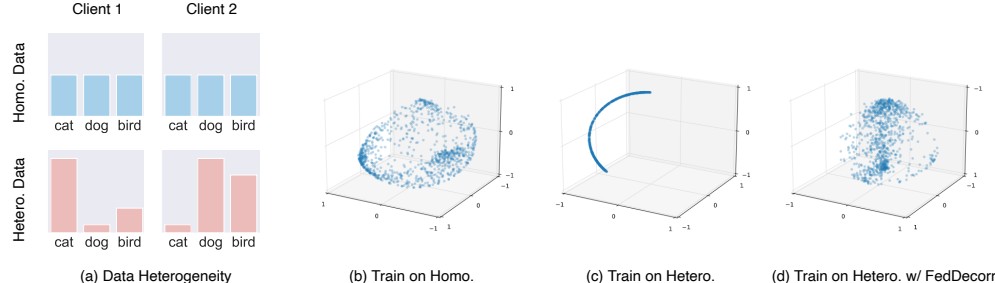

Figure 1: **(a)** illustrates data heterogeneity in terms of number of samples per class. **(b)**, **(c)**, **(d)** show representations (normalized to the unit sphere) of global models trained under homogeneous data, heterogeneous data, and heterogeneous data with FEDDECORR, respectively. Only **(c)** suffers dimensional collapse. **(b)**, **(c)**, **(d)** are produced with ResNet20 on CIFAR10. Best viewed in color.

study the representations output by each global model. Interestingly, we find that as the degree of data heterogeneity increases, more singular values tend to evolve towards zero. This observation suggests that stronger data heterogeneity causes the trained global model to suffer from more severe *dimensional collapse*, whereby representations are biased towards residing in a lower-dimensional space (or manifold). A graphical illustration of how heterogeneous training data affect output representations is shown in Fig. 1(b-c). Our observations suggest that dimensional collapse might be one of the key reasons why federated learning methods struggle under data heterogeneity. Essentially, dimensional collapse is a form of *oversimplification* in terms of the model, where the representation space is not being fully utilized to discriminate diverse data of different classes.

Given the observations made on the global model, we conjecture that the dimensional collapse of the global model is inherited from models locally trained on various clients. This is because the global model is a result of the aggregation of local models. To validate our conjecture, we further visualize the local models in terms of the singular values of representation covariance matrices in Sec. 3.2. Similar to the visualization on the global model, we observe dimensional collapse on representations produced by local models. With this observation, we establish the connection between dimensional collapse of the global model and local models. To further understand the dimensional collapse on local models, we analyze the gradient flow dynamics of local training in Sec. 3.3. Interestingly, we show theoretically that heterogeneous data drive the weight matrices of the local models to be biased to being low-rank, which further results in representation dimensional collapse.

Inspired by the observations that dimensional collapse of the global model stems from local models, we consider mitigating dimensional collapse during local training in Sec. 4. In particular, we propose a novel federated learning method termed FEDDECORR. FEDDECORR adds a regularization term during local training to encourage the Frobenius norm of the correlation matrix of representations to be small. We show theoretically and empirically that this proposed regularization term can effectively mitigate dimensional collapse (see Fig. 1(d) for example). Next, in Sec. 5, through extensive experiments on standard benchmark datasets including CIFAR10, CIFAR100, and TinyImageNet, we show that FEDDECORR consistently improves over baseline federated learning methods. In addition, we find that FEDDECORR yields more dramatic improvements in more challenging federated learning setups such as stronger heterogeneity or more number of clients. Lastly, FEDDECORR has extremely low computation overhead and can be built on top of any existing federated learning baseline methods, which makes it widely applicable.

Our contributions are summarized as follows. First, we discover through experiments that stronger data heterogeneity in federated learning leads to greater dimensional collapse for global and local models. Second, we develop a theoretical understanding of the dynamics behind our empirical discovery that connects data heterogeneity and dimensional collapse. Third, based on the motivation of mitigating dimensional collapse, we propose a novel method called FEDDECORR, which yields consistent improvements while being implementation-friendly and computationally-efficient.

## 2 RELATED WORKS

**Federated Learning.** McMahan et al. (2017) proposed FedAvg, which adopts a simple averaging scheme to aggregate local models into the global model. However, under data heterogeneity, FedAvg

suffers from unstable and slow convergence, resulting in performance degradation. To tackle this challenge, previous works either improve local training (Li et al., 2021b; 2020; Karimireddy et al., 2020; Acar et al., 2021; Al-Shedivat et al., 2020; Wang et al., 2021) or global aggregation (Wang et al., 2020b; Hsu et al., 2019; Luo et al., 2021; Wang et al., 2020a; Lin et al., 2020; Reddi et al., 2020; Wang et al., 2020a). Most of these methods focus on the model parameter space, which may result in high computation or communication cost due to deep neural networks being over-parameterized. Li et al. (2021b) focuses on model representations and uses a contrastive loss to maximize agreements between representations of local models and the global model. However, one drawback of Li et al. (2021b) is that it requires additional forward passes during training, which almost doubles the training cost. In this work, based on our study of how data heterogeneity affects model representations, we propose an effective yet highly efficient method to handle heterogeneous data. Another research trend is in personalized federated learning (Arivazhagan et al., 2019; Li et al., 2021c; Fallah et al., 2020; T Dinh et al., 2020; Hanzely et al., 2020; Huang et al., 2021; Zhang et al., 2020), which aims to train personalized local models for each client. In this work, however, we focus on the typical setting that aims to train one global model for all clients.

**Dimensional Collapse.** Dimensional collapse of representations has been studied in metric learning (Roth et al., 2020), self-supervised learning (Jing et al., 2021), and class incremental learning (Shi et al., 2022). In this work, we focus on federated learning and discover that stronger data heterogeneity causes a higher degree of dimensional collapse for locally trained models. To the best of our knowledge, this work is the first to discover and analyze dimensional collapse of representations in federated learning.

**Gradient Flow Dynamics.** Arora et al. (2018; 2019) introduce the gradient flow dynamics framework to analyze the dynamics of multi-layer linear neural networks under the $\ell_2$-loss and find deeper neural networks biasing towards low-rank solution during optimization. Following their works, Jing et al. (2021) finds two factors that cause dimensional collapse in self-supervised learning, namely strong data augmentation and implicit regularization from depth. Differently, we focus on federated learning with the cross-entropy loss. More importantly, our analysis focuses on dimensional collapse caused by data heterogeneity in federated learning instead of depth of neural networks.

**Feature Decorrelation.** Feature decorrelation had been used for different purposes, such as preventing mode collapse in self-supervised learning (Bardes et al., 2021; Zbontar et al., 2021; Hua et al., 2021), boosting generalization (Cogswell et al., 2015; Huang et al., 2018; Xiong et al., 2016), and improving class incremental learning (Shi et al., 2022). We instead apply feature decorrelation to counter the undesired dimensional collapse caused by data heterogeneity in federated learning.

## 3 DIMENSIONAL COLLAPSE CAUSED BY DATA HETEROGENEITY

In this section, we first empirically visualize and compare representations of global models trained under different degrees of data heterogeneity in Sec. 3.1. Next, to better understand the observations on global models, we analyze representations of local models in Sec. 3.2. Finally, to theoretically understand our observations, we analyze the gradient flow dynamics of local training in Sec. 3.3.

### 3.1 EMPIRICAL OBSERVATIONS ON THE GLOBAL MODEL

We first empirically demonstrate that stronger data heterogeneity causes more severe dimensional collapse on the global model. Specifically, we first separate the training samples of CIFAR100 into 10 splits, each corresponding to the local data of one client. To simulate data heterogeneity among clients as in previous works (Yurochkin et al., 2019; Wang et al., 2020a; Li et al., 2021b), we sample a probability vector $\mathbf{p}_c = (p_{c,1}, p_{c,2}, \ldots, p_{c,K}) \sim \text{Dir}_K(\alpha)$ and allocate a $p_{c,k}$ proportion of instances of class $c \in [C] = \{1, 2, \ldots, C\}$ to client $k \in [K]$, where $\text{Dir}_K(\alpha)$ is the Dirichlet distribution with $K$ categories and $\alpha$ is the concentration parameter. A smaller $\alpha$ implies stronger data heterogeneity ($\alpha = \infty$ corresponds to the homogeneous setting). We let $\alpha \in \{0.01, 0.05, 0.25, \infty\}$.

For each of the settings generated by different $\alpha$'s, we apply FedAvg (McMahan et al., 2017) to train a MobileNetV2 (Sandler et al., 2018) with CIFAR100 (observations on other federated learning methods, model architectures, or datasets are similar and are provided in Appendix D). Next, for each of the four trained global models, we compute the covariance matrix $\Sigma = \frac{1}{N} \sum_{i=1}^{N} (\mathbf{z}_i -$

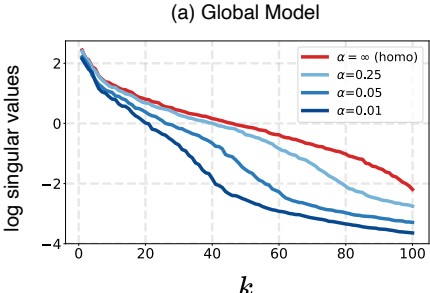 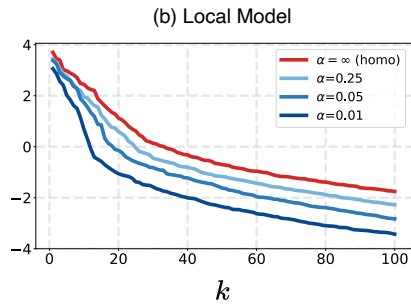

Figure 2: Data heterogeneity causes dimensional collapse on **(a) global models** and **(b) local models.** We plot the singular values of the covariance matrix of representations in descending order. The $x$-axis ($k$) is the index of singular values and the $y$-axis is the logarithm of the singular values.

$\bar{\mathbf{z}})(\mathbf{z}_i - \bar{\mathbf{z}})^\top$ of the representations over the $N$ test data points in CIFAR100. Here $\mathbf{z}_i$ is the $i$-th test data point and $\bar{\mathbf{z}} = \frac{1}{N} \sum_{i=1}^{N} \mathbf{z}_i$ is their average.

Finally, we apply the singular value decomposition (SVD) on each of the covariance matrices and visualize the top 100 singular values in Fig. 2(a). If we define a small value $\tau$ as the threshold for a singular value to be *significant* (e.g., $\log \tau = -2$), we observe that for the homogeneous setting, almost all the singular values are significant, i.e., they surpass $\tau$. However, as $\alpha$ decreases, the number of singular values exceeding $\tau$ monotonically decreases. This implies that with stronger heterogeneity among local training data, the representation vectors produced by the trained global model tend to reside in a lower-dimensional space, corresponding to more severe dimensional collapse.

## 3.2 Empirical Observations on Local Models

Since the global model is obtained by aggregating locally trained models on each client, we conjecture that the dimensional collapse observed on the global model stems from the dimensional collapse of local models. To further validate our conjecture, we continue to study whether increasing data heterogeneity will also lead to more severe dimensional collapse on locally trained models.

Specifically, for different $\alpha$'s, we visualize the locally trained model of one client (visualizations on local models of other clients are similar and are provided in Appendix E). Following the same procedure as in Sec. 3.1, we plot the singular values of covariance matrices of representations produced by the local models. We observe from Fig. 2(b) that locally trained models demonstrate the same trend as the global models—namely, that the presence of stronger data heterogeneity causes more severe dimensional collapse. These experiments corroborate that the global model inherit the adverse dimensional collapse phenomenon from the local models.

## 3.3 A Theoretical Explanation for Dimensional Collapse

Based on the empirical observations in Sec. 3.1 and Sec. 3.2, we now develop a theoretical understanding to explain why heterogeneous training data causes dimensional collapse for the learned representations.

Since we have established that the dimensional collapse of global model stems from local models, we focus on studying local models in this section. Without loss of generality, we study local training of one arbitrary client. Specifically, we first analyze the gradient flow dynamics of the model weights during the local training. This analysis shows how heterogeneous local training data drives the model weights towards being low-rank, which leads to dimensional collapse for the representations.

### 3.3.1 Setups and Notations

We denote the number of training samples as $N$, the dimension of input data as $d_{\text{in}}$, and total number of classes as $C$. The $i$-th sample is denoted as $X_i \in \mathbb{R}^{d_{\text{in}}}$, and its corresponding one-hot encoded label is $\mathbf{y}_i \in \mathbb{R}^C$. The collection of all $N$ training samples is denoted as $X = [X_1, X_2 \ldots, X_N] \in \mathbb{R}^{d_{\text{in}} \times N}$, and the $N$ one-hot encoded training labels are denoted as $\mathbf{y} = [\mathbf{y}_1, \mathbf{y}_2, \ldots, \mathbf{y}_N] \in \mathbb{R}^{C \times N}$.

For simplicity in exposition, we follow Arora et al. (2018; 2019) and Jing et al. (2021) and analyze linear neural networks (without nonlinear activation layers). We consider an $(L + 1)$-layer (where $L \geq 1$) linear neural network trained using the cross entropy loss under gradient flow (i.e., gradient descent with an infinitesimally small learning rate). The weight matrix of the $i$-th layer ($i \in [L+1]$) at the optimization time step $t$ is denoted as $W_i(t)$. The dynamics can be expressed as

$$\dot{W}_i(t) = -\frac{\partial}{\partial W_i}\ell(W_1(t), \ldots, W_{L+1}(t)), \tag{1}$$

where $\ell$ denotes the cross-entropy loss.

In addition, at the optimization time step $t$ and given the input data $X_i$, we denote $\mathbf{z}_i(t) \in \mathbb{R}^d$ as the output representation vector ($d$ being the dimension of the representations) and $\gamma_i(t) \in \mathbb{R}^C$ as the output softmax probability vector. We have

$$\gamma_i(t) = \text{softmax}(W_{L+1}(t)\mathbf{z}_i(t)) = \text{softmax}(W_{L+1}(t)W_L(t)\ldots W_1(t)X_i). \tag{2}$$

We define $\mu_c = \frac{N_c}{N}$, where $N_c$ is number of data samples belonging to class $c$. We denote $\mathbf{e}_c$ as the $C$-dimensional one-hot vector where only the $c$-th entry is 1 (and the others are 0). In addition, let $\bar{\gamma}_c(t) = \frac{1}{N_c}\sum_{i=1}^N \gamma_i(t)\mathbb{1}\{\mathbf{y}_i = \mathbf{e}_c\}$ and $\bar{X}_c = \frac{1}{N_c}\sum_{i=1}^N X_i\mathbb{1}\{\mathbf{y}_i = \mathbf{e}_c\}$.

### 3.3.2 ANALYSIS ON GRADIENT FLOW DYNAMICS

Since our goal is to analyze model representations $\mathbf{z}_i(t)$, we focus on weight matrices that directly produce representations (i.e., the first $L$ layers). We denote the product of the weight matrices of the first $L$ layers as $\Pi(t) = W_L(t)W_{L-1}(t)\ldots W_1(t)$ and analyze the behavior of $\Pi(t)$ under the gradient flow dynamics. In particular, we derive the following result for the singular values of $\Pi(t)$.

**Theorem 1** (Informal). *Assuming that the mild conditions as stated in Appendix A.3 hold. Let $\sigma_k(t)$ for $k \in [d]$ be the $k$-th largest singular value of $\Pi(t)$. Then,*

$$\dot{\sigma}_k(t) = NL\,(\sigma_k(t))^{2-\frac{2}{L}}\sqrt{(\sigma_k(t))^{\frac{2}{L}} + M}\,(\mathbf{u}_{L+1,k}(t))^{\top}G(t)\mathbf{v}_k(t), \tag{3}$$

*where $\mathbf{u}_{L+1,k}(t)$ is the $k$-th left singular vector of $W_{L+1}(t)$, $\mathbf{v}_k(t)$ is the $k$-th right singular vector of $\Pi(t)$, $M$ is a constant, and $G(t)$ is defined as*

$$G(t) = \sum_{c=1}^C \mu_c(\mathbf{e}_c - \bar{\gamma}_c(t))\bar{X}_c^{\top}, \tag{4}$$

*where $\mu_c$, $\mathbf{e}_c$, $\bar{\gamma}_c(t)$, $\bar{X}_c$ are defined after Eqn. (2).*

The proof of the precise version of Theorem 1 is provided in Appendix A.

Based on Theorem 1, we are able to explain why greater data heterogeneity causes $\Pi(t)$ to be biased to become lower-rank. Note that strong data heterogeneity causes local training data of one client being highly imbalanced in terms of the number of data samples per class (recall Fig. 1(a)). This implies that $\mu_c$, which is the proportion of the class $c$ data, will be close to 0 for some classes.

Next, based on the definition of $G(t)$ in Eqn. (4), more $\mu_c$'s being close to 0 leads to $G(t)$ being biased towards a low-rank matrix. If this is so, the term $(\mathbf{u}_{L+1,k}(t))^{\top}G(t)\mathbf{v}_k(t)$ in Eqn. (3) will only be significant (large in magnitude) for fewer values of $k$. This is because $\mathbf{u}_{L+1,k}(t)$ and $\mathbf{v}_k(t)$ are both singular vectors, which are orthogonal among different $k$'s. This further leads to $\dot{\sigma}_k(t)$ on the left-hand side of Eqn. (3), which is the evolving rate of $\sigma_k$, being small for most of the $k$'s throughout training. These observations imply that only relatively few singular values of $\Pi(t)$ will increase significantly after training.

Furthermore, $\Pi(t)$ being biased towards being low-rank will directly lead to dimensional collapse for the representations. To see this, we simply write the covariance matrix of the representations in terms of $\Pi(t)$ as

$$\Sigma(t) = \frac{1}{N}\sum_{i=1}^N (\mathbf{z}_i(t) - \bar{\mathbf{z}}(t))(\mathbf{z}_i(t) - \bar{\mathbf{z}}(t))^{\top} = \Pi(t)\left(\frac{1}{N}\sum_{i=1}^N (X_i - \bar{X})(X_i - \bar{X})^{\top}\right)\Pi(t)^{\top}. \tag{5}$$

From Eqn. (5), we observe that if $\Pi(t)$ evolves to being a lower-rank matrix, $\Sigma(t)$ will also tend to be lower-rank, which corresponds to the stronger dimensional collapse observed in Fig. 2(b).

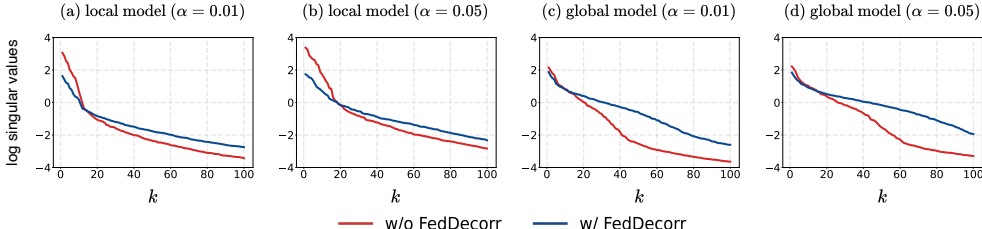

Figure 3: FEDDECORR effectively mitigates dimensional collapse for **(a-b) local models and (c-d) global models**. For each heterogeneity parameter $\alpha \in \{0.01, 0.05\}$, we apply FEDDECORR and plot the singular values of the representation covariance matrix. The $x$-axis ($k$) is the index of singular values. With FEDDECORR, the tail singular values are prevented from dropping to 0 too rapidly.

## 4 MITIGATING DIMENSIONAL COLLAPSE WITH FEDDECORR

Motivated by the above observations and analyses on dimensional collapse caused by data heterogeneity in federated learning, we explore how to mitigate excessive dimensional collapse.

Since dimensional collapse on the global model is inherited from local models, we propose to alleviate the problem during local training. One natural way to achieve this is to add the following regularization term on the representations during training

$$L_{\text{singular}}(w, X) = \frac{1}{d} \sum_{i=1}^{d} \left( \lambda_i - \frac{1}{d} \sum_{j=1}^{d} \lambda_j \right)^2, \tag{6}$$

where $\lambda_i$ is the $i$-th singular value of the covariance matrix of the representations. Essentially, $L_{\text{singular}}$ penalizes the variance among the singular values, thus discouraging the tail singular values from collapsing to 0, mitigating dimensional collapse. However, this regularization term is not practical as it requires calculating all the singular values, which is computationally expensive.

Therefore, to derive a computationally-cheap training objective, we first apply the z-score normalization on all the representation vectors $\mathbf{z}_i$ as follows: $\hat{\mathbf{z}}_i = (\mathbf{z}_i - \bar{\mathbf{z}})/\sqrt{\text{Var}(\mathbf{z})}$. This results in the covariance matrix of $\hat{\mathbf{z}}_i$ being equal to its correlation matrix (i.e., the matrix of correlation coefficients). The following proposition suggests a more convenient cost function to regularize.

**Proposition 1.** *For a $d$-by-$d$ correlation matrix $K$ with singular values $(\lambda_1, \ldots, \lambda_d)$, we have:*

$$\sum_{i=1}^{d} \left( \lambda_i - \frac{1}{d} \sum_{j=1}^{d} \lambda_j \right)^2 = \|K\|_{\text{F}}^2 - d. \tag{7}$$

The proof of Proposition 1 can be found in Appendix B. This proposition suggests that regularizing the Frobenius norm of the correlation matrix $\|K\|_{\text{F}}$ achieves the same effect as minimizing $L_{\text{singular}}$. In contrast to the singular values, $\|K\|_{\text{F}}$ can be computed efficiently.

To leverage this proposition, we propose a novel method, FEDDECORR, which regularizes the Frobenius norm of the correlation matrix of the representation vectors during local training on each client. Formally, the proposed regularization term is defined as:

$$L_{\text{FedDecorr}}(w, X) = \frac{1}{d^2} \|K\|_{\text{F}}^2, \tag{8}$$

where $w$ is the model parameters, $K$ is the correlation matrix of the representations. The overall objective of each local client is

$$\min_{w} \ell(w, X, \mathbf{y}) + \beta L_{\text{FedDecorr}}(w, X), \tag{9}$$

where $\ell$ is the cross entropy loss, and $\beta$ is the regularization coefficient of FEDDECORR. The pseudocode of our method is provided in Appendix G.

To visualize the effectiveness of $L_{\text{FedDecorr}}$ in mitigating dimensional collapse, we now revisit the experiments of Fig. 2 and apply $L_{\text{FedDecorr}}$ under the heterogeneous setting where $\alpha \in \{0.01, 0.05\}$. We plot our results in Fig. 3 for both local and global models. Figs. 3(a-b) show that for local models, FEDDECORR encourages the tail singular values to not collapse to 0, thus effectively mitigating dimensional collapse. Moreover, as illustrated in Figs. 3(c-d), this desirable effect introduced by FEDDECORR on local models can also be inherited by the global models.

| Method | CIFAR10 | | | | CIFAR100 | | | |
|---|---|---|---|---|---|---|---|---|
| | $\alpha = 0.05$ | 0.1 | 0.5 | $\infty$ | 0.05 | 0.1 | 0.5 | $\infty$ |
| FedAvg | $64.85_{\pm2.01}$ | $76.28_{\pm1.22}$ | $89.84_{\pm0.13}$ | $92.39_{\pm0.26}$ | $59.87_{\pm0.25}$ | $66.46_{\pm0.16}$ | $71.69_{\pm0.15}$ | $\mathbf{74.54}_{\pm0.15}$ |
| + FEDDECORR | $73.06_{\pm0.81}$ | $80.60_{\pm0.91}$ | $89.84_{\pm0.05}$ | $92.19_{\pm0.10}$ | $\mathbf{61.53}_{\pm0.11}$ | $\mathbf{67.12}_{\pm0.09}$ | $\mathbf{71.91}_{\pm0.04}$ | $73.87_{\pm0.18}$ |
| FedProx | $64.11_{\pm0.84}$ | $76.10_{\pm0.40}$ | $89.57_{\pm0.04}$ | $92.38_{\pm0.09}$ | $60.02_{\pm0.46}$ | $66.41_{\pm0.27}$ | $71.78_{\pm0.19}$ | $74.34_{\pm0.03}$ |
| + FEDDECORR | $71.38_{\pm0.81}$ | $\mathbf{81.74}_{\pm0.34}$ | $89.96_{\pm0.26}$ | $92.14_{\pm0.20}$ | $61.33_{\pm0.19}$ | $67.00_{\pm0.46}$ | $71.64_{\pm0.10}$ | $74.15_{\pm0.06}$ |
| FedAvgM | $71.34_{\pm0.71}$ | $77.51_{\pm0.58}$ | $88.39_{\pm0.17}$ | $91.35_{\pm0.15}$ | $59.64_{\pm0.20}$ | $66.36_{\pm0.14}$ | $71.17_{\pm0.22}$ | $74.20_{\pm0.16}$ |
| + FEDDECORR | $\mathbf{73.60}_{\pm0.82}$ | $79.21_{\pm0.15}$ | $88.70_{\pm0.26}$ | $91.33_{\pm0.13}$ | $61.48_{\pm0.27}$ | $66.60_{\pm0.11}$ | $71.26_{\pm0.21}$ | $73.86_{\pm0.25}$ |
| MOON | $68.79_{\pm0.69}$ | $78.70_{\pm0.66}$ | $90.08_{\pm0.10}$ | $92.62_{\pm0.17}$ | $56.79_{\pm0.17}$ | $65.48_{\pm0.29}$ | $71.81_{\pm0.14}$ | $74.30_{\pm0.12}$ |
| + FEDDECORR | $73.46_{\pm0.84}$ | $81.63_{\pm0.55}$ | $\mathbf{90.61}_{\pm0.05}$ | $\mathbf{92.63}_{\pm0.19}$ | $59.43_{\pm0.34}$ | $66.12_{\pm0.20}$ | $71.68_{\pm0.05}$ | $73.70_{\pm0.25}$ |

Table 1: **CIFAR10/100 Experiments.** We run experiments under various degrees of heterogeneity ($\alpha \in \{0.05, 0.1, 0.5, \infty\}$) and report the test accuracy (%). All results are (re)produced by us and are averaged over 3 runs (mean $\pm$ std). Bold font highlights the highest accuracy in each column.

# 5 EXPERIMENTS

## 5.1 EXPERIMENTAL SETUPS

**Datasets:** We adopt three popular benchmark datasets, namely CIFAR10, CIFAR100, and TinyImageNet. CIFAR10 and CIFAR100 both have $50,000$ training samples and $10,000$ test samples, and the size of each image is $32 \times 32$. TinyImageNet contains 200 classes, with $100,000$ training samples and $10,000$ testing samples, and each image is $64 \times 64$. The method generating local data for each client was introduced in Sec. 3.1.

| Method | TinyImageNet | | | |
|---|---|---|---|---|
| | $\alpha = 0.05$ | 0.1 | 0.5 | $\infty$ |
| FedAvg | $35.02_{\pm0.46}$ | $39.30_{\pm0.23}$ | $46.92_{\pm0.25}$ | $49.33_{\pm0.19}$ |
| + FEDDECORR | $40.29_{\pm0.18}$ | $43.86_{\pm0.50}$ | $50.01_{\pm0.27}$ | $52.63_{\pm0.26}$ |
| FedProx | $35.20_{\pm0.30}$ | $39.66_{\pm0.43}$ | $47.16_{\pm0.07}$ | $49.76_{\pm0.36}$ |
| + FEDDECORR | $\mathbf{40.63}_{\pm0.05}$ | $44.19_{\pm0.14}$ | $50.26_{\pm0.27}$ | $52.37_{\pm0.36}$ |
| FedAvgM | $34.81_{\pm0.09}$ | $39.72_{\pm0.11}$ | $47.11_{\pm0.04}$ | $49.67_{\pm0.25}$ |
| + FEDDECORR | $39.97_{\pm0.23}$ | $43.95_{\pm0.26}$ | $50.14_{\pm0.11}$ | $52.05_{\pm0.37}$ |
| MOON | $35.23_{\pm0.26}$ | $40.53_{\pm0.28}$ | $47.25_{\pm0.66}$ | $50.48_{\pm0.57}$ |
| + FEDDECORR | $40.40_{\pm0.24}$ | $\mathbf{44.20}_{\pm0.22}$ | $\mathbf{50.81}_{\pm0.51}$ | $\mathbf{53.01}_{\pm0.45}$ |

Table 2: **TinyImageNet Experiments.** We run with $\alpha \in \{0.05, 0.1, 0.5, \infty\}$) and report the test accuracy (%). All results are (re)produced by us and are averaged over 3 runs (mean $\pm$ std is reported). Bold font highlights the highest accuracy in each column.

**Implementation Details:** Our code is based on the code of Li et al. (2021b). For all experiments, we use MobileNetV2 (Sandler et al., 2018). We run 100 communication rounds for all experiments on the CIFAR10/100 datasets and 50 communication rounds on the TinyImageNet dataset. We conduct local training for 10 epochs in each communication round using SGD optimizer with a learning rate of 0.01, a SGD momentum of 0.9, and a batch size of 64. The weight decay is set to $10^{-5}$ for CIFAR10 and $10^{-4}$ for CIFAR100 and TinyImageNet. We apply the data augmentation of Cubuk et al. (2018) in all CIFAR100 and TinyImageNet experiments. The $\beta$ of FEDDECORR (i.e., $\beta$ in Eqn. (9)) is tuned to be 0.1. The details of tuning hyper-parameters for other federated learning methods are described in Appendix F.

## 5.2 FEDDECORR SIGNIFICANTLY IMPROVES BASELINE METHODS

To validate the effectiveness of our method, we apply FEDDECORR to four baselines, namely FedAvg (McMahan et al., 2017), FedAvgM (Hsu et al., 2019), FedProx (Li et al., 2020), and MOON (Li et al., 2021b). We partition the three benchmark datasets (CIFAR10, CIFAR100, and TinyImageNet) into 10 clients with $\alpha \in \{0.05, 0.1, 0.5, \infty\}$. Since $\alpha = \infty$ is the homogeneous setting where local models should be free from the pitfall of excessive dimensional collapse, we only expect FEDDECORR to perform on par with the baselines in this setting.

We display the CIFAR10/100 results in Tab. 1 and the TinyImageNet results in Tab. 2. We observe that for all of the heterogeneous settings on all datasets, the highest accuracies are achieved

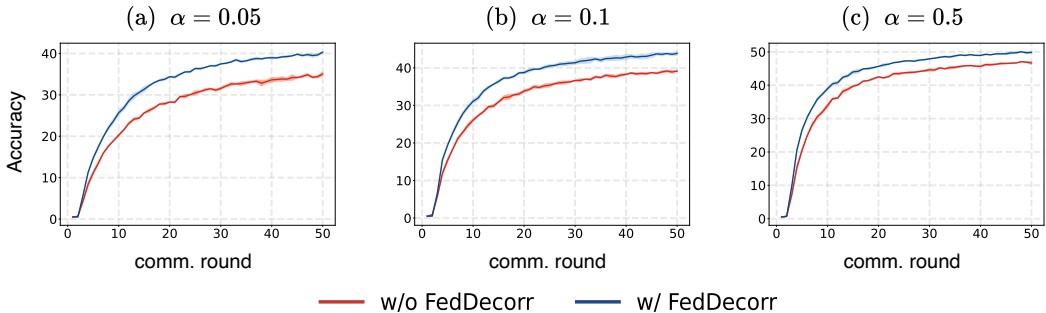

Figure 4: **Test accuracy (%) at each communication round.** Results are averaged over 3 runs. Shaded areas denote one standard deviation above and below the mean.

by adding FEDDECORR on top of a certain baseline method. In particular, in the strongly heterogeneous settings where $\alpha \in \{0.05, 0.1\}$, adding FEDDECORR yields significant improvements of around $\mathbf{2\% \sim 9\%}$ over baseline methods on all datasets. On the other hand, for the less heterogeneous setting of $\alpha = 0.5$, the problem of dimensional collapse is less pronounced as discussed in Sec 3, leading to smaller improvements from FEDDECORR. Such decrease in improvements is a general trend and is also observed on FedProx, FedAvgM, and MOON. In addition, surprisingly, in the homogeneous setting of $\alpha = \infty$, FEDDECORR still produces around 2% of improvements on the TinyImageNet dataset. We conjecture that this is because TinyImageNet is much more complicated than the CIFAR datasets, and some other factors besides heterogeneity of label may cause undesirable dimensional collapse in the federated learning setup. Therefore, federated learning on TinyImageNet can benefit from FEDDECORR even in the homogeneous setting.

To further demonstrate the advantages of FED-DECORR, we apply it on FedAvg and plot how the test accuracy of the global model evolves throughout the federated learning in Fig. 4. In this figure, if we set a certain value of the testing accuracy as a threshold, we see that adding FEDDECORR significantly reduces the number of communication rounds needed to achieve the given threshold. This further shows that FED-DECORR not only improves the final performance, but also greatly boosts the communication efficiency in federated learning.

| # clients | Method | $\alpha = 0.05$ | 0.1 | 0.5 |
|---|---|---|---|---|
| 10 | FedAvg | 35.02 | 39.30 | 46.92 |
| | + FEDDECORR | 40.29 | 43.86 | 50.01 |
| 20 | FedAvg | 31.21 | 35.30 | 43.64 |
| | + FEDDECORR | 39.41 | 41.27 | 46.17 |
| 30 | FedAvg | 26.20 | 30.88 | 37.22 |
| | + FEDDECORR | 36.50 | 39.02 | 44.38 |
| 50 | FedAvg | 25.70 | 28.88 | 34.89 |
| | + FEDDECORR | 34.50 | 36.67 | 42.34 |
| 100 | FedAvg | 21.53 | 24.69 | 30.21 |
| | + FEDDECORR | 30.55 | 33.85 | 38.65 |

Table 3: **Ablation study on the number of clients.** Based on TinyImageNet, we run experiments with different number of clients and different amounts of data heterogeneity.

## 5.3 ABLATION STUDY ON THE NUMBER OF CLIENTS

Next, we study whether the improvements brought by FEDDECORR are preserved as number of clients increases. We partition the TinyImageNet dataset into 10, 20, 30, 50, and 100 clients according to different $\alpha$'s, and then run FedAvg with and without FEDDECORR. For the experiments with 10, 20 and 30 clients, we run 50 communication rounds. For the experiments with 50 and 100 clients, we randomly select 20% of the total clients to participate the federated learning in each round and run 100 communication rounds. Results are shown in Tab. 3. From this table, we see that the performance improvements resulting from FEDDECORR increase from around $\mathbf{3\% \sim 5\%}$ to around $\mathbf{7\% \sim 10\%}$ with the growth in the number of clients. Therefore, interestingly, we show through experiments that the improvements brought by FEDDECORR can be even more pronounced under the more challenging settings with more clients. Moreover, our experimental results under random client participation show that the improvements from FEDDECORR are robust to such uncertainties. These experiments demonstrate the potential of FEDDECORR to be applied to real world federated learning settings with massive numbers of clients and random client participation.

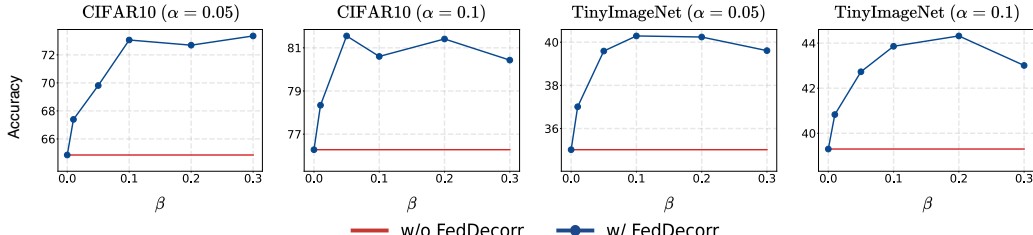

Figure 5: **Ablation study on** $\beta$**.** We apply FEDDECORR with different choices of $\beta$ on FedAvg.

## 5.4 ABLATION STUDY ON THE REGULARIZATION COEFFICIENT $\beta$

Next, we study FEDDECORR's robustness to the $\beta$ in Eqn. (9) by varying it in the set $\{0.01, 0.05, 0.1, 0.2, 0.3\}$. We partition the CIFAR10 and TinyImageNet datasets into 10 clients with $\alpha$ equals to 0.05 and 0.1 to simulate the heterogeneous setting. Results are shown in Fig. 5. We observe that, in general, when $\beta$ increases, the performance of FEDDECORR first increases, then plateaus, and finally decreases slightly. These results show that FEDDECORR is relatively insensitive to the choice of $\beta$, which implies FEDDECORR is an easy-to-tune federated learning method. In addition, among all experimental setups, setting $\beta$ to be 0.1 consistently produces (almost) the best results. Therefore, we recommend $\beta = 0.1$ when having no prior information about the dataset.

## 5.5 ABLATION STUDY ON THE NUMBER OF LOCAL EPOCHS

Lastly, we ablate on the number of local epochs per communication round. We set the number of local epochs $E$ to be in the set $\{1, 5, 10, 20\}$. We run experiments with and without FEDDECORR, and we use the CIFAR100 and TinyImageNet datasets with $\alpha$ being 0.05 and 0.1 for this ablation study. Results are shown in Tab. 4, in which one observes that with increasing $E$, FEDAVG performance first increases and then decreases. This is because when $E$ is too small, the local training cannot converge properly in each communication round. On the other hand, when $E$ is too large, the model parameters of local clients might be driven to be too far from the global optimum. Nevertheless, FEDDECORR consistently improves over the baselines across different choices of local epochs $E$.

| $E$ | Method | CIFAR100 | | TinyImageNet | |
|---|---|---|---|---|---|
| | | $\alpha = 0.05$ | 0.1 | 0.05 | 0.1 |
| 1 | FedAvg | 50.67 | 55.98 | 32.31 | 34.88 |
| | + FEDDECORR | 53.18 | 57.02 | 36.49 | 38.99 |
| 5 | FedAvg | 59.57 | 65.02 | 36.02 | 40.75 |
| | + FEDDECORR | 61.42 | 65.98 | 41.68 | 44.77 |
| 10 | FedAvg | 59.87 | 66.46 | 35.02 | 39.30 |
| | + FEDDECORR | 61.53 | 67.12 | 40.29 | 43.86 |
| 20 | FedAvg | 58.50 | 66.37 | 31.23 | 37.23 |
| | + FEDDECORR | 60.65 | 66.86 | 35.44 | 42.04 |

Table 4: **Ablation study on local epochs.** Experiments with different number of local epochs $E$.

## 5.6 ADDITIONAL EMPIRICAL ANALYSES

We present more empirical analyses in Appendix C. These include comparing FEDDECORR with other baselines (Appendix C.4) and other decorrelation methods (Appendix C.2), experiments on other model architectures (Appendix C.3) and another type of data heterogeneity (Appendix C.5), and discussing the computational advantage of FEDDECORR (Appendix C.1).

## 6 CONCLUSION

In this work, we study representations of trained models under federated learning in which the data held by clients are heterogeneous. Through extensive empirical observations and theoretical analyses, we show that stronger data heterogeneity results in more severe dimensional collapse for both global and local representations. Motivated by this, we propose FEDDECORR, a novel method to mitigate dimensional collapse during local training, thus improving federated learning under the heterogeneous data setting. Extensive experiments on benchmark datasets show that FEDDECORR yields consistent improvements over existing baseline methods.

ACKNOWLEDGEMENTS

The authors would like to thank anonymous reviewers for the constructive feedback. Yujun Shi and Vincent Tan are supported by Singapore Ministry of Education Tier 1 grants (Grant Number: A-0009042-01-00, A-8000189-01-00, A-8000980-00-00) and a Singapore National Research Foundation (NRF) Fellowship (Grant Number: A-0005077-01-00). Jian Liang is supported by National Natural Science Foundation of China (Grant No. 62276256) and Beijing Nova Program under Grant Z211100002121108.

## REPRODUCIBILITY STATEMENT

All source code has been released at https://github.com/bytedance/FedDecorr. Pseudo-code of FED-DECORR is provided in Appendix G. We introduced all the implementation details of baselines and our method in Sec. 5.1. In addition, the proofs of Theorem 1 and Proposition 1 are provided in Appendix A and Appendix B, respectively. All assumptions are stated and discussed in the proof.

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

# A PROOF OF THEOREM 1 IN MAIN PAPER

## A.1 NOTATIONS REVISITED

Here, for the reader's convenience, we summarize the notations used in both the main text and this appendix.

| Notation | Explanation |
|---|---|
| $N$ | Number of training data points. |
| $C$ | Total number of classes. |
| $X$ | The collection of the $N$ training samples, $X \in \mathbb{R}^{d_{\text{in}} \times N}$. |
| $\mathbf{y}$ | The collection of one hot labels of the $N$ training samples, $\mathbf{y} \in \mathbb{R}^{C \times N}$. |
| $\gamma$ | The collection of model output softmax vectors given all $N$ input data, $\gamma \in \mathbb{R}^{C \times N}$ |
| $W_i(t)$ | The $i$-th layer weight matrix at the $t$-th optimization step. |
| $\Pi(t)$ | The product of the weight matrices of the first $L$ layers: $\Pi(t) = W_L(t) \dots W_1(t)$. |
| $\sigma_{l,k}$ | The $k$-th singular value of $W_l$. |
| $\mathbf{u}_{l,k}$ | The $k$-th left singular vector of $W_l$. |
| $\mathbf{v}_{l,k}$ | The $k$-th right singular vector of $W_l$. |
| $\sigma_k$ | The $k$-th singular value of $\Pi$. |
| $\mathbf{u}_k$ | The $k$-th left singular vector of $\Pi$. |
| $\mathbf{v}_k$ | The $k$-th right singular vector of $\Pi$. |
| $N_c$ | Number of samples of class $c$. |
| $\mu_c$ | The proportion of class $c$ samples w.r.t. the whole training samples: $\mu_c = \frac{N_c}{N}$ |
| $\mathbf{e}_c$ | The $C$-dimensional one-hot vector where only the $c$-th entry is 1. |
| $\bar{X}_c$ | Mean vector of the training examples in class $c$: $\bar{X}_c = \frac{1}{N_c} \sum_{i=1}^{N} X_i \mathbb{1}\{\mathbf{y}_i = \mathbf{e}_c\}$ |
| $\bar{\gamma}_c$ | Mean output softmax vector given samples in class $c$: $\bar{\gamma}_c = \frac{1}{N_c} \sum_{i=1}^{N} \gamma_i \mathbb{1}\{\mathbf{y}_i = \mathbf{e}_c\}$ |

## A.2 TWO LEMMAS

Here, we elaborate two useful lemmas from Arora et al. (2019; 2018).

The first lemma is adopted from Arora et al. (2019):

**Lemma 1.** *Assuming the weight matrix $W$ evolves under gradient descent dynamics with infinitesimally small learning rate, the $k$-th singular value of this matrix (denoted as $\sigma_k$) evolves as*

$$\dot{\sigma}_k(t) = (\mathbf{u}_k(t))^\top \dot{W}(t) \mathbf{v}_k(t), \tag{10}$$

*where $\mathbf{u}_k(t)$ and $\mathbf{v}_k(t)$ are the $k$-th left and right singular vectors of $W(t)$, respectively.*

*Proof.* By performing an SVD on $W(t)$, we have $W(t) = U(t)S(t)V(t)^\top$. Therefore, by the chain rule in differention, we have:

$$\dot{W}(t) = \dot{U}(t)S(t)V(t)^\top + U(t)\dot{S}(t)V(t)^\top + U(t)S(t)\dot{V}(t)^\top. \tag{11}$$

Next, for both sides of the above equation, we left multiply $U(t)^\top$ and right multiply $V(t)$:

$$U(t)^\top \dot{W}(t) V(t) = U(t)^\top \dot{U}(t) S(t) + \dot{S}(t) + S(t)(\dot{V}(t))^\top V(t). \tag{12}$$

Since $S(t)$ is a diagonal matrix, we consider the $k$-th diagonal entry of $S(t)$, namely $\sigma_k(t)$:

$$(\mathbf{u}_k(t))^\top \dot{W}(t) \mathbf{v}_k(t) = (\mathbf{u}_k(t))^\top \dot{\mathbf{u}}_k(t) \sigma_k(t) + \dot{\sigma}_k(t) + \sigma_k(t)(\mathbf{v}_k(t))^\top \dot{\mathbf{v}}_k(t). \tag{13}$$

Since $\mathbf{u}_k(t)$ and $\mathbf{v}_k(t)$ are unit vectors and are evolving in time with infinitesimal rate, we have $(\mathbf{u}_k(t))^\top \dot{\mathbf{u}}_k(t) = 0$ and $(\mathbf{v}_k(t))^\top \dot{\mathbf{v}}_k(t) = 0$. Next, Eqn. (13) can be simplified as

$$\dot{\sigma}_k(t) = (\mathbf{u}_k(t))^\top \dot{W}(t) \mathbf{v}_k(t). \tag{14}$$

The proof is thus complete. □

The second lemma is adopted from Arora et al. (2018).

**Lemma 2.** *Given $L$ consecutive linear layers in a neural network characterized by weight matrices $W_1, W_2, \ldots, W_L$. We denote $\Pi = W_L W_{L-1} \ldots W_1$. We further denote $W_j(t)$ as weight matrix $W_j$ after the $t$-th gradient descent optimization step. Correspondingly, the initialization of $W_j$ is $W_j(0)$. Assuming we have $W_j(0)(W_j(0))^\top = (W_{j+1}(0))^\top W_{j+1}(0)$ for any $j \in [L-1]$ at initialization. Then, under the gradient descent dynamics, $\Pi(t)$ satisfies*

$$\dot{\Pi}(t) = -\sum_{j=1}^{L} \left[ \Pi(t)\Pi(t)^\top \right]^{\frac{L-j}{L}} \frac{\partial \ell(\Pi(t))}{\partial \Pi} \left[ \Pi(t)^\top \Pi(t) \right]^{\frac{j-1}{L}}, \tag{15}$$

*where $[\cdot]^{\frac{L-j}{L}}$ and $[\cdot]^{\frac{j-1}{L}}$ are fractional power operators defined over positive semi-definite matrices.*

*Proof.* Here, we first define some additional notation. Given any square matrices (or possibly scalar) $A_1, A_2, \ldots, A_m$, we denote $\mathrm{diag}(A_1, A_2, \ldots, A_m)$ to be the block diagonal matrix

$$\mathrm{diag}(A_1, A_2, \ldots, A_m) = \begin{bmatrix} A_1 & 0 & 0 & 0 \\ 0 & A_2 & 0 & 0 \\ 0 & 0 & \ddots & 0 \\ 0 & 0 & 0 & A_m \end{bmatrix}.$$

Here, we first consider dynamics of an arbitrary $W_j$ where $j \in [L-1]$. By the chain rule, we have

$$\dot{W}_j(t) = -\frac{\partial \ell(W_1(t), \ldots, W_{L+1}(t))}{\partial W_j(t)}$$
$$= -(W_{j+1}(t)^\top \ldots W_L(t)^\top) \frac{\partial \ell(\Pi(t))}{\partial \Pi} (W_1(t)^\top \ldots W_{j-1}(t)^\top). \tag{16}$$

Given Eqn. (16), we right multiply $\dot{W}_j(t)$ by $(W_j(t))^\top$ and we left multiply $\dot{W}_{j+1}(t)$ by $(W_{j+1}(t))^\top$, which yields

$$\dot{W}_j(t)(W_j(t))^\top = (W_{j+1}(t))^\top \dot{W}_{j+1}(t). \tag{17}$$

Applying the same trick on $W_j(t)^\top$ and $W_{j+1}(t)^\top$ yields

$$W_j(t)(\dot{W}_j(t))^\top = (\dot{W}_{j+1}(t))^\top W_{j+1}(t). \tag{18}$$

Adding Eqns. (17) and (18) on both sides yields

$$\dot{W}_j(t)(W_j(t))^\top + W_j(t)(\dot{W}_j(t))^\top = W_{j+1}(t)^\top \dot{W}_{j+1}(t) + (\dot{W}_{j+1}(t))^\top W_{j+1}(t). \tag{19}$$

Next, by the chain rule for differentiation, Eqn. (19) directly implies that

$$\frac{\mathrm{d}(W_j(t)W_j(t)^\top)}{\mathrm{d}t} = \frac{\mathrm{d}(W_{j+1}(t)^\top W_{j+1}(t))}{\mathrm{d}t}. \tag{20}$$

Since we have assumed that $W_j(0)W_j(0)^\top = W_{j+1}(0)^\top W_{j+1}(0)$, we can conclude that

$$W_j(t)W_j(t)^\top = W_{j+1}(t)^\top W_{j+1}(t). \tag{21}$$

Next, we apply an SVD on $W_j(t)$ and $W_{j+1}(t)$ in Eqn. (21). This yields

$$U_j(t)S_j(t)S_j^\top(t)U_j^\top(t) = V_{j+1}(t)S_{j+1}^\top(t)S_{j+1}(t)V_{j+1}^\top(t). \tag{22}$$

Based on Eqn. (22) and given the uniqueness property of SVD, we know:

$$S_j(t)S_j(t)^\top = S_{j+1}^\top(t)S_{j+1}(t) = \mathrm{diag}(\rho_1 I_{d_1}, \rho_2 I_{d_2}, \ldots, \rho_m I_{d_m}), \tag{23}$$

where $\sqrt{\rho_1}, \ldots, \sqrt{\rho_m}$ represent the $m$ distinct singular values satisfying $\rho_1 > \rho_2 > \ldots > \rho_m \geq 0$, and $I_{d_r}$ for any $r \in [m]$ are identity matrix of size $d_r \times d_r$. Since Eqn. (23) holds for any $j$, we know by induction that the set of values of $\rho$'s is the same across all layers $j \in [L]$. In addition, there exist orthogonal matrices $O_{j,r} \in \mathbb{R}^{d_r \times d_r}$ for any $r \in [m]$ such that

$$U_j(t) = V_{j+1}(t)\mathrm{diag}(O_{j,1}, O_{j,2}, \ldots, O_{j,m}). \tag{24}$$

Given Eqns. (24), next, we study $W_{j+1}(t)W_j(t)W_j^\top(t)W_{j+1}^\top(t)$ for any $j \in [N-1]$:

$$
\begin{aligned}
&W_{j+1}(t)W_j(t)W_j^\top(t)W_{j+1}^\top(t)\\
&= U_{j+1}S_{j+1}V_{j+1}^\top U_j S_j S_j^\top U_j^\top V_{j+1}S_{j+1}^\top U_{j+1}^\top\\
&= U_{j+1}S_{j+1}\mathrm{diag}(O_{j,1}, O_{j,2}, \ldots, O_{j,m})S_j S_j^\top \mathrm{diag}(O_{j,1}^\top, O_{j,2}^\top, \ldots, O_{j,m}^\top)S_{j+1}^\top U_{j+1}^\top\\
&\hspace{8cm}\text{(plugging-in (24))}\\
&= U_{j+1}S_{j+1}S_j S_j^\top S_{j+1}^\top U_{j+1}^\top \qquad (S_j \text{ commutes with } \mathrm{diag}(O_{j,1}, O_{j,2}, \ldots, O_{j,m}))\\
&= U_{j+1}\mathrm{diag}(\rho_1^2 I_{d_1}, \rho_2^2 I_{d_2}, \ldots, \rho_m^2 I_{d_m})U_{j+1}^\top.
\end{aligned}
\tag{25}
$$

Similarly, it holds that

$$
W_j^\top(t)W_{j+1}^\top(t)W_{j+1}(t)W_j(t) = V_j\mathrm{diag}(\rho_1^2 I_{d_1}, \rho_2^2 I_{d_2}, \ldots, \rho_m^2 I_{d_m})V_j^\top.
\tag{26}
$$

Next, by induction and Eqns. (25),

$$
\begin{aligned}
&W_L(t)\ldots W_j(t)W_j(t)^\top \ldots W_L(t)^\top\\
&\qquad = U_L\mathrm{diag}(\rho_1^{L-j+1} I_{d_1}, \rho_2^{L-j+1} I_{d_2}, \ldots, \rho_m^{L-j+1} I_{d_m})U_L^\top,
\end{aligned}
\tag{27}
$$

by induction and Eqns. (26), it holds that

$$
W_1^\top(t)\ldots W_j^\top(t)W_j(t)\ldots W_1(t) = V_1\mathrm{diag}(\rho_1^j I_{d_1}, \rho_2^j I_{d_2}, \ldots, \rho_m^j I_{d_m})V_1^\top.
\tag{28}
$$

From Eqns. (27), we know that for any $j \in [L-1]$,

$$
\begin{aligned}
\Pi(t)\Pi(t)^\top &= W_L(t)\ldots W_1(t)W_1(t)^\top \ldots W_L(t)^\top\\
&= U_L\mathrm{diag}(\rho_1^L I_{d_1}, \rho_2^L I_{d_2}, \ldots, \rho_m^L I_{d_m})U_L^\top\\
&= \left[U_L\mathrm{diag}(\rho_1^{L-j} I_{d_1}, \rho_2^{L-j} I_{d_2}, \ldots, \rho_m^{L-j} I_{d_m})U_L^\top\right]^{\frac{L}{L-j}}\\
&= \left[W_L(t)\ldots W_{j+1}(t)W_{j+1}(t)^\top \ldots W_L(t)^\top\right]^{\frac{L}{L-j}}.
\end{aligned}
\tag{29}
$$

Similarly, from Eqn. (28), we know that for any $2 \le j \le L-1$,

$$
\begin{aligned}
\Pi(t)^\top\Pi(t) &= W_1(t)^\top \ldots W_L(t)^\top W_L(t)\ldots W_1(t)\\
&= V_1\mathrm{diag}(\rho_1^L I_{d_1}, \rho_2^L I_{d_2}, \ldots, \rho_m^L I_{d_m})V_1^\top\\
&= \left[V_1\mathrm{diag}(\rho_1^{j-1} I_{d_1}, \rho_2^{j-1} I_{d_2}, \ldots, \rho_m^{j-1} I_{d_m})V_1^\top\right]^{\frac{L}{j-1}}\\
&= \left[W_1^\top \ldots W_{j-1}^\top W_{j-1}(t)\ldots W_1(t)\right]^{\frac{L}{j-1}}.
\end{aligned}
\tag{30}
$$

With everything derived above, we now study the dynamics of $\Pi(t)$ as follows

$$
\begin{aligned}
\dot\Pi(t) &= \sum_{j=1}^{L}\left[W_L(t)\ldots W_{j+1}(t)\right](\dot W_j(t))\left[W_{j-1}(t)\ldots W_1(t)\right] \quad \text{(differential chain rule)}\\
&= -\sum_{j=1}^{L}\left[W_L(t)\ldots W_{j+1}(t)W_{j+1}(t)^\top \ldots W_L(t)^\top\right]\\
&\quad \times \frac{\partial \ell(\Pi(t))}{\partial \Pi}\left[W_1^\top(t)\ldots W_{j-1}^\top(t)W_{j-1}(t)\ldots W_1(t)\right] \quad \text{(plugging-in (16))}\\
&= -\sum_{j=1}^{L}\left[\Pi(t)\Pi(t)^\top\right]^{\frac{L-j}{L}}\frac{\partial \ell(\Pi(t))}{\partial \Pi}\left[\Pi(t)^\top\Pi(t)\right]^{\frac{j-1}{L}} \quad \text{(plugging-in (29) and (30))}.
\end{aligned}
\tag{31}
$$

This completes the proof. $\qquad\qquad\square$

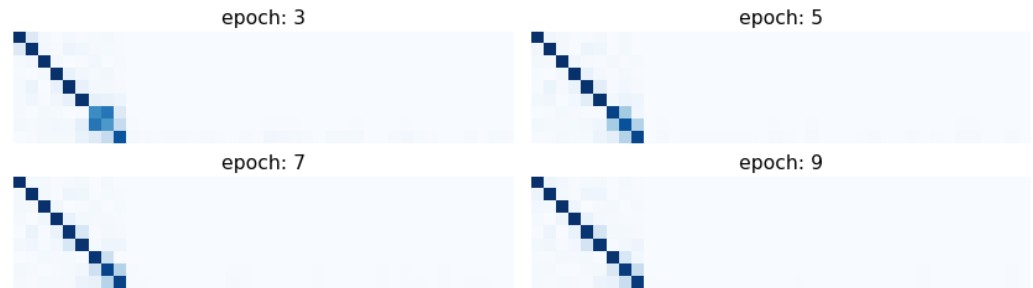

Figure 6: **Alignment effects between the singular spaces of $W_{L+1}(t)$ and $\Pi(t)$.** We train a 3-layer linear neural network on the MNIST dataset and visualize the models at 3, 5, 7, 9 training epochs, respectively. In each figure, the $k'$-th row and $k$-th column pixel is the value of $|\mathbf{u}_k(t)^\top \mathbf{v}_{L+1,k'}(t)|$. Darker colors denote values close to 1 while lighter colors denote values close to 0. From the figures, we empirically observe that $|\mathbf{u}_k(t)^\top \mathbf{v}_{L+1,k'}(t)| = \mathbb{1}\{k = k'\}$ approximately holds.

### A.3  ASSUMPTIONS

**Assumption 1.** *We assume that the initial values of the weight matrices satisfy $W_{i+1}^\top(0)W_{i+1}(0) = W_i(0)W_i^\top(0)$ for any $i \in [L-1]$.*

**Assumption 2.** *We assume $|\mathbf{u}_k(t)^\top \mathbf{v}_{L+1,k'}(t)| = \mathbb{1}\{k = k'\}$ holds for all $t$, where $\mathbf{u}_k(t)$ is the $k$-th left singular vector of $\Pi(t)$ and $\mathbf{v}_{L+1,k'}(t)$ is the $k'$-th right singular vector of $W_{L+1}(t)$.*

**Remark:** For Assumption 1, it can be achieved in practice by proper random initialization. For Assumption 2, Ji & Telgarsky (2018) proved that under some assumptions, gradient descent optimization will drive consecutive layers of linear networks to satisfy it. We also provide empirical evidence in Fig. 6 to corroborate that this assumption approximately holds.

### A.4  PROOF OF THEOREM 1 IN THE MAIN TEXT

**Theorem 1 (formally stated).** *Let $\sigma_k(t)$ for $k \in [d]$ be the $k$-th largest singular value of $\Pi(t)$. Then, under Assumptions 1 and 2, we have*

$$\dot{\sigma}_k(t) = NL \, (\sigma_k(t))^{2-\frac{2}{L}} \sqrt{(\sigma_k(t))^{\frac{2}{L}} + M} \, (\mathbf{u}_{L+1,k}(t))^\top G(t)\mathbf{v}_k(t), \tag{32}$$

*where $\mathbf{u}_{L+1,k}(t)$ is the $k$-th left singular vector of $W_{L+1}(t)$, $\mathbf{v}_k(t)$ is the $k$-th right singular vector of $\Pi(t)$, $M$ is a constant, and $G(t)$ is defined as*

$$G(t) = \sum_{c=1}^{C} \mu_c(\mathbf{e}_c - \bar{\gamma}_c(t))\bar{X}_c^\top. \tag{33}$$

*Proof.* Recall that for $(L+1)$-layer linear neural networks, given the $i$-th training sample $X_i \in \mathbb{R}^d$, we have

$$\gamma_i(t) = \text{softmax}(W_{L+1}(t)\mathbf{z}_i(t)) = \text{softmax}(W_{L+1}(t)\Pi(t)X_i), \tag{34}$$

and the loss is the standard cross-entropy loss defined as follows

$$\ell(\Pi(t), W_{L+1}(t)) = \sum_{i=1}^{N} -\mathbf{y}_i^\top \log \gamma_i(t). \tag{35}$$

By the chain rule, we can derive gradient of $\ell$ with respect to $W_{L+1}$ and $\Pi$, which are respectively,

$$\frac{\partial \ell(\Pi(t), W_{L+1}(t))}{\partial W_{L+1}} = -(\mathbf{y} - \gamma(t))X^\top \Pi(t)^\top, \tag{36}$$

and

$$\frac{\partial \ell(\Pi(t), W_{L+1}(t))}{\partial \Pi} = -W_{L+1}(t)^\top(\mathbf{y} - \gamma(t))X^\top. \tag{37}$$

Next, under the gradient descent dynamics, the dynamics on $W_{L+1}$ satisfies

$$\dot{W}_{L+1}(t) = -\frac{\partial \ell(\Pi(t), W_{L+1}(t))}{\partial W_{L+1}} = (\mathbf{y} - \gamma(t))X^\top \Pi(t)^\top, \tag{38}$$

while the dynamics on $\Pi$ requires invoking Lemma 2, which allows us to write

$$\dot{\Pi}(t) = -\sum_{j=1}^{L} [\Pi(t)\Pi(t)^\top]^{\frac{L-j}{L}} \frac{\partial \ell(\Pi(t))}{\partial \Pi} [\Pi(t)^\top \Pi(t)]^{\frac{j-1}{L}}$$

$$= \sum_{j=1}^{L} [\Pi(t)\Pi(t)^\top]^{\frac{L-j}{L}} W_{L+1}(t)^\top (\mathbf{y} - \gamma(t))X^\top [\Pi(t)^\top \Pi(t)]^{\frac{j-1}{L}}. \tag{39}$$

Next, we invoke Lemma 1 on Eqn. (39) and Eqn. (38), respectively, yielding:

$$\dot{\sigma}_k(t) = (\mathbf{u}_k(t))^\top \dot{\Pi}(t)(\mathbf{v}_k(t))$$

$$= \sum_{j=1}^{L} \mathbf{u}_k(t)^\top [\Pi(t)\Pi(t)^\top]^{\frac{L-j}{L}} W_{L+1}(t)^\top (\mathbf{y} - \gamma(t))X^\top [\Pi(t)^\top \Pi(t)]^{\frac{j-1}{L}} \mathbf{v}_k(t)$$

$$= L(\sigma_k(t))^{2-\frac{2}{L}} \mathbf{u}_k(t)^\top W_{L+1}(t)^\top (\mathbf{y} - \gamma(t))X^\top \mathbf{v}_k(t)$$

$$\text{(SVD on } \Pi(t))$$

$$= L(\sigma_k(t))^{2-\frac{2}{L}} \sum_{k'} \sigma_{L+1,k'} \mathbf{u}_k(t)^\top \mathbf{v}_{L+1,k'}(t)(\mathbf{u}_{L+1,k'}(t))^\top (\mathbf{y} - \gamma(t))X^\top \mathbf{v}_k(t)$$

$$\text{(SVD on } W_{L+1}(t))$$

$$= L(\sigma_k(t))^{2-\frac{2}{L}} \sigma_{L+1,k}(\mathbf{u}_{L+1,k}(t))^\top (\mathbf{y} - \gamma(t))X^\top \mathbf{v}_k(t) \quad \text{(Assumption 2).}$$

$$\tag{40}$$

and

$$\dot{\sigma}_{L+1,k}(t) = \mathbf{u}_{L+1,k}(t)^\top (\mathbf{y} - \gamma(t))X^\top \Pi(t)^\top \mathbf{v}_{L+1,k}(t)$$

$$= \sum_{k'} \sigma_{k'} \mathbf{u}_{L+1,k}(t)^\top (\mathbf{y} - \gamma(t))X^\top \mathbf{v}_{k'}(t) \mathbf{u}_{k'}^\top \mathbf{v}_{L+1,k}(t) \tag{41}$$

$$= \sigma_k \mathbf{u}_{L+1,k}(t)^\top (\mathbf{y} - \gamma(t))X^\top \mathbf{v}_k(t) \quad \text{(Assumption 2).}$$

Combining Eqns. (40) and (41), we have:

$$\frac{1}{L}(\dot{\sigma}_k(t))(\sigma_k(t))^{\frac{2}{L}-1} = \sigma_{L+1,k}(t)(\dot{\sigma}_{L+1,k}(t)). \tag{42}$$

Next, apply integration on both sides, which yields

$$(\sigma_{L+1,k}(t))^2 = (\sigma_k(t))^{\frac{2}{L}} + M, \tag{43}$$

where $M$ a constant.

By Eqn. (43), Eqn. (40) can be rewritten as

$$\dot{\sigma}_k(t) = L(\sigma_k(t))^{2-\frac{2}{L}} \sqrt{(\sigma_k(t))^{\frac{2}{L}} + M} \, (\mathbf{u}_{L+1,k}(t))^\top (\mathbf{y} - \gamma(t))X^\top \mathbf{v}_k(t). \tag{44}$$

Finally, notice that $(\mathbf{y} - \gamma(t))X^\top$ can be rewritten as

$$(\mathbf{y} - \gamma(t))X^\top = \sum_{i=1}^{N} (\mathbf{y} - \gamma_i(t))X_i^\top = N \sum_{c=1}^{C} \mu_c(\mathbf{e}_c - \bar{\gamma}_c(t))\bar{X}_c^\top. \tag{45}$$

We further substitute Eqn. (45) into Eqn. (44) and obtain

$$\dot{\sigma}_k(t) = NL(\sigma_k(t))^{2-\frac{2}{L}} \sqrt{(\sigma_k(t))^{\frac{2}{L}} + M} \, (\mathbf{u}_{L+1,k}(t))^\top G(t)\mathbf{v}_k(t), \tag{46}$$

where $G(t)$ is defined as

$$G(t) = \sum_{c=1}^{C} \mu_c(\mathbf{e}_c - \bar{\gamma}_c(t))\bar{X}_c^\top. \tag{47}$$

This completes the proof. $\qquad \square$

## B PROOF OF PROPOSITION 1 IN THE MAIN PAPER

**Proposition 1 (restated).** *For a $d$-by-$d$ correlation matrix $K$ with singular values $(\lambda_1, \ldots, \lambda_d)$, we have:*

$$\sum_{i=1}^{d} \left( \lambda_i - \frac{1}{d} \sum_{j=1}^{d} \lambda_j \right)^2 = \|K\|_{\mathrm{F}}^2 - d. \tag{48}$$

*Proof.* Given a $d$-by-$d$ correlation matrix $K$, since the diagonal entries of $K$ are all 1, we have

$$\sum_{i=1}^{d} \lambda_i = \mathrm{tr}(K) = d. \tag{49}$$

This is because for any symmetric positive definite matrix, the sum of all singular values equals the trace of the matrix.

Next, for the left-hand side of Eqn. (7), we have:

$$\sum_{i=1}^{d} \left( \lambda_i - \frac{1}{d} \sum_{j=1}^{d} \lambda_j \right)^2 = \sum_{i=1}^{d} (\lambda_i - 1)^2 \qquad \text{(Plug-in Eqn. (49))}$$

$$= \sum_{i=1}^{d} \lambda_i^2 - 2 \sum_{i=1}^{d} \lambda_i + d \tag{50}$$

$$= \sum_{i=1}^{d} \lambda_i^2 - d \qquad \text{(Plug-in Eqn. (49))}.$$

Next, for the right-hand side of Eqn. (7), we have:

$$\|K\|_{\mathrm{F}}^2 - d = \mathrm{tr}(K^\top K) - d$$

$$= \mathrm{tr}(U S V^\top V S^\top U^\top) - d \qquad \text{(Apply SVD on } K\text{)}$$

$$= \mathrm{tr}(U S S^\top U^\top) - d \tag{51}$$

$$= \sum_{i=1}^{n} \lambda_i^2 - d.$$

Therefore, we have shown that the left-hand side of Eqn. (7) equals its right-hand side. $\square$

## C ADDITIONAL EMPIRICAL ANALYSES

### C.1 COMPUTATIONAL EFFICIENCY

We demonstrate FEDDECORR's advantage vis-à-vis some of its competitors in terms of its computational efficiency. We compare FEDDECORR with some other methods that also apply additional regularization terms during local training such as FedProx and MOON. We partition CIFAR10, CIFAR100 and TinyImageNet into 10 clients with $\alpha = 0.5$ and report the total computation times required for one round of training for FedAvg, FedProx, MOON, and FEDDECORR . Results are shown in Tab. 5. All results are produced with a NVIDIA Tesla V100 GPU. We see that FEDDECORR incurs a negligible computation overhead on top of the naïve FedAvg, while FedProx and MOON introduce about $0.5 \sim 1\times$ additional computation cost. The advantage of FEDDECORR in terms of efficiency is mainly because it only involves calculating the Frobenius norm of a matrix which is extremely cheap. Indeed this regularization operates on the output representation vectors of the model, without requiring computing parameter-wise regularization like FedProx nor extra forward passes like MOON.

|  | CIFAR10 | CIFAR100 | TinyImageNet |
|---|---|---|---|
| FedAvg | 6.7 | 6.9 | 25.4 |
| FedProx | 12.1 | 12.3 | 33.2 |
| MOON | 12.2 | 12.7 | 38.1 |
| FEDDECORR | 6.9 | 7.1 | 25.7 |

Table 5: **Comparison of computation times.** We report the total computation times (in minutes) for one round of training on the three datasets for FedAvg, FedProx, MOON, and FEDDECORR. Here, FEDDECORR stands for applying FEDDECORR to FedAvg.

## C.2 COMPARISON WITH OTHER DECORRELATION METHODS

Some decorrelation regularizations such as DeCov (Cogswell et al., 2015) and Structured-DeCov (Xiong et al., 2016) were proposed to improve the generalization capabilities in standard classification tasks. Both these methods operate directly on the covariance matrix of the representations instead of the correlation matrix like our proposed method—FEDDECORR. To compare our FEDDECORR with the existing decorrelation methods, we follow the same procedure as in FEDDECORR and apply DeCov and Structured-DeCov during local training. Our experiments are based on TinyImageNet and FedAvg. TinyImageNet is partitioned into 10 clients according to various $\alpha$'s. Results are shown in Tab. 6. Surprisingly, we see that unlike our FEDDECORR which steadily improves the baseline, adding DeCov or Structured-DeCov both degrade the performance in federated learning. We conjecture that this is because directly regularizing the covariance matrix is highly unstable, leading to undesired modification on the representations. This experiment shows that our design of regularization of the *correlation matrix* instead of the *covariance matrix* is of paramount importance.

|  | FedAvg | DeCov | St.-Decov | FEDDECORR |
|---|---|---|---|---|
| $\alpha = 0.05$ | 35.02 | 32.88 | 32.04 | 40.29 |
| $\alpha = 0.1$ | 39.30 | 37.29 | 37.74 | 43.86 |
| $\alpha = 0.5$ | 46.92 | 46.29 | 45.85 | 50.01 |

Table 6: **Comparison with other decorrelation methods.** Based on FedAvg and the TinyImageNet dataset, we use different decorrelation regularizers in local training.

## C.3 EXPERIMENTS ON OTHER MODEL ARCHITECTURES

In this section, we demonstrate the effectiveness of our method across different model architectures. Here, besides the MobileNetV2 used in the main paper, we also experiment on ResNet18 and ResNet32. Note that ResNet18 is the wider ResNet whose representation dimension is $512$ and ResNet32 is the narrower ResNet whose representation dimension is $64$. The coefficient of the FedDecorr objective is set to be $0.1$ as suggested to be a good universal value of $\beta$ in the paper. The heterogeneity parameter $\alpha$ is set to be $0.05$ and we use the CIFAR10 dataset. Our results are shown in Tab. 7. As can be seen, FedDecorr yields consistent improvements across different neural network architectures. One interesting phenomenon is that the improvements brought about by FedDecorr are much larger on wider networks (e.g., MobileNetV2, ResNet18) than on narrower ones (e.g. ResNet32). We conjecture this is because the dimension of the ambient space of wider networks are clearly higher than that of shallower networks. Therefore, relatively speaking, the dimensional collapse caused by data heterogeneity will be more severe for wider networks.

|  | MobileNetV2 | ResNet18 | ResNet32 |
|---|---|---|---|
| FedAvg | 64.85 | 71.51 | 65.76 |
| + FEDDECORR | 73.06 | 76.54 | 67.21 |

Table 7: **Effectiveness of FEDDECORR on other model architectures.**

| Method | CIFAR10 | | | | CIFAR100 | | | |
|---|---|---|---|---|---|---|---|---|
| | $\alpha = 0.05$ | 0.1 | 0.5 | $\infty$ | 0.05 | 0.1 | 0.5 | $\infty$ |
| Scaffold | $51.99_{\pm 2.54}$ | $74.36_{\pm 3.10}$ | $87.05_{\pm 0.39}$ | $89.77_{\pm 0.24}$ | $54.51_{\pm 0.26}$ | $61.42_{\pm 0.54}$ | $68.37_{\pm 0.44}$ | $70.97_{\pm 0.04}$ |
| FedNova | $63.07_{\pm 1.59}$ | $79.98_{\pm 1.56}$ | $90.23_{\pm 0.41}$ | $92.39_{\pm 0.18}$ | $60.22_{\pm 0.33}$ | $66.43_{\pm 0.26}$ | $71.79_{\pm 0.17}$ | $74.47_{\pm 0.13}$ |
| FedAvg | $64.85_{\pm 2.01}$ | $76.28_{\pm 1.22}$ | $89.84_{\pm 0.13}$ | $92.39_{\pm 0.26}$ | $59.87_{\pm 0.25}$ | $66.46_{\pm 0.16}$ | $71.69_{\pm 0.15}$ | $\mathbf{74.54}_{\pm 0.15}$ |
| + FedDecorr | $73.06_{\pm 0.81}$ | $80.60_{\pm 0.91}$ | $89.84_{\pm 0.05}$ | $92.19_{\pm 0.10}$ | $\mathbf{61.53}_{\pm 0.11}$ | $\mathbf{67.12}_{\pm 0.09}$ | $\mathbf{71.91}_{\pm 0.04}$ | $73.87_{\pm 0.18}$ |
| FedProx | $64.11_{\pm 0.84}$ | $76.10_{\pm 0.40}$ | $89.57_{\pm 0.04}$ | $92.38_{\pm 0.09}$ | $60.02_{\pm 0.46}$ | $66.41_{\pm 0.27}$ | $71.78_{\pm 0.19}$ | $74.34_{\pm 0.03}$ |
| + FedDecorr | $71.38_{\pm 0.81}$ | $\mathbf{81.74}_{\pm 0.34}$ | $89.96_{\pm 0.26}$ | $92.14_{\pm 0.20}$ | $61.33_{\pm 0.19}$ | $67.00_{\pm 0.46}$ | $71.64_{\pm 0.10}$ | $74.15_{\pm 0.06}$ |
| FedAvgM | $71.34_{\pm 0.71}$ | $77.51_{\pm 0.58}$ | $88.39_{\pm 0.17}$ | $91.35_{\pm 0.15}$ | $59.64_{\pm 0.20}$ | $66.36_{\pm 0.14}$ | $71.17_{\pm 0.22}$ | $74.20_{\pm 0.16}$ |
| + FedDecorr | $\mathbf{73.60}_{\pm 0.82}$ | $79.21_{\pm 0.15}$ | $88.70_{\pm 0.26}$ | $91.33_{\pm 0.13}$ | $61.48_{\pm 0.27}$ | $66.60_{\pm 0.11}$ | $71.26_{\pm 0.21}$ | $73.86_{\pm 0.25}$ |
| MOON | $68.79_{\pm 0.69}$ | $78.70_{\pm 0.66}$ | $90.08_{\pm 0.10}$ | $92.62_{\pm 0.17}$ | $56.79_{\pm 0.17}$ | $65.48_{\pm 0.29}$ | $71.81_{\pm 0.14}$ | $74.30_{\pm 0.12}$ |
| + FedDecorr | $73.46_{\pm 0.84}$ | $81.63_{\pm 0.55}$ | $\mathbf{90.61}_{\pm 0.05}$ | $\mathbf{92.63}_{\pm 0.19}$ | $59.43_{\pm 0.34}$ | $66.12_{\pm 0.20}$ | $71.68_{\pm 0.05}$ | $73.70_{\pm 0.25}$ |

Table 8: **CIFAR10/100 Experiments.** We run experiments under various degrees of heterogeneity ($\alpha \in \{0.05, 0.1, 0.5, \infty\}$) and report the test accuracy (%). All results are (re)produced by us and are averaged over 3 runs (mean $\pm$ std). Bold font highlights the highest accuracy in each column. We add results of Scaffold and FedNova comparing to Tab. 1 in the main paper.

| Method | TinyImageNet | | | |
|---|---|---|---|---|
| | $\alpha = 0.05$ | 0.1 | 0.5 | $\infty$ |
| Scaffold | $35.16_{\pm 0.77}$ | $37.87_{\pm 0.78}$ | $44.24_{\pm 0.14}$ | $44.88_{\pm 0.29}$ |
| FedNova | $35.28_{\pm 0.04}$ | $39.73_{\pm 0.07}$ | $47.05_{\pm 0.42}$ | $49.57_{\pm 0.09}$ |
| FedAvg | $35.02_{\pm 0.46}$ | $39.30_{\pm 0.23}$ | $46.92_{\pm 0.25}$ | $49.33_{\pm 0.19}$ |
| + FedDecorr | $40.29_{\pm 0.18}$ | $43.86_{\pm 0.50}$ | $50.01_{\pm 0.27}$ | $52.63_{\pm 0.26}$ |
| FedProx | $35.20_{\pm 0.30}$ | $39.66_{\pm 0.43}$ | $47.16_{\pm 0.07}$ | $49.76_{\pm 0.36}$ |
| + FedDecorr | $\mathbf{40.63}_{\pm 0.05}$ | $44.19_{\pm 0.14}$ | $50.26_{\pm 0.27}$ | $52.37_{\pm 0.36}$ |
| FedAvgM | $34.81_{\pm 0.09}$ | $39.72_{\pm 0.11}$ | $47.11_{\pm 0.04}$ | $49.67_{\pm 0.25}$ |
| + FedDecorr | $39.97_{\pm 0.23}$ | $43.95_{\pm 0.26}$ | $50.14_{\pm 0.11}$ | $52.05_{\pm 0.37}$ |
| MOON | $35.23_{\pm 0.26}$ | $40.53_{\pm 0.28}$ | $47.25_{\pm 0.66}$ | $50.48_{\pm 0.57}$ |
| + FedDecorr | $40.40_{\pm 0.24}$ | $\mathbf{44.20}_{\pm 0.22}$ | $\mathbf{50.81}_{\pm 0.51}$ | $\mathbf{53.01}_{\pm 0.45}$ |

Table 9: **TinyImageNet Experiments.** We run with $\alpha \in \{0.05, 0.1, 0.5, \infty\}$ and report the test accuracies (%). All results are (re)produced by us and are averaged over 3 runs (mean $\pm$ std is reported). Bold font highlights the highest accuracy in each column. We add results of Scaffold and FedNova comparing to Tab. 2 in the main paper.

## C.4 Comparison with Other Federated Learning Baselines

In this section, we compare FedDecorr with two other baselines, namely Scaffold (Karimireddy et al., 2020) and FedNova (Wang et al., 2020b). We use the same experimental setups as in the main paper to implement these two baselines. Results on CIFAR10/100 and TinyImageNet are shown in Tab. 8 and Tab. 9, respectively. As shown in the tables, across various datasets and degrees of heterogeneity, adding FedDecorr on top of a baseline method can outperform the baselines when there is some heterogeneity across the agents, i.e., $\alpha < \infty$.

## C.5 Experiments on Another Type of Heterogeneity

In this section, we run experiments under another type of data heterogeneity. Specifically, we follow McMahan et al. (2017) and split the CIFAR10 dataset across different clients such that each client only has a fixed number of classes $C$ (e.g., $C = 2$ indicates each client only has data of two classes). We split the data across 10 clients and choose $C$ to be 2 and 3. Results are shown in Tab. 10. As can

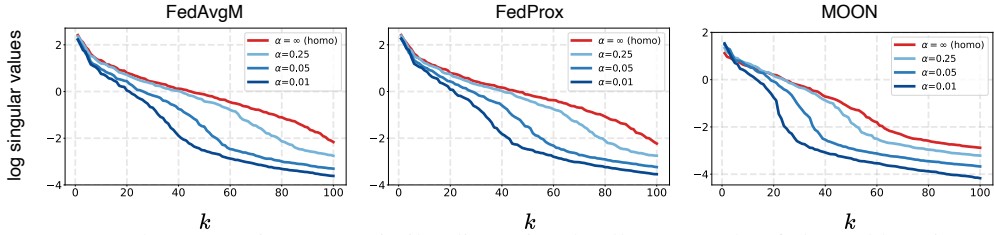

Figure 7: Data heterogeneity causes similar dimensional collapse on other federated learning methods such as FedAvgM (Hsu et al., 2019), FedProx (Li et al., 2020), and MOON (Li et al., 2021b). The x-axis ($k$) is the index of singular values.

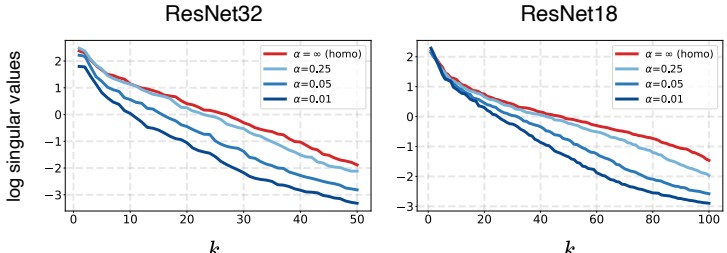

Figure 8: Data heterogeneity causes similar dimensional collapse on other model architectures during federated learning. The x-axis ($k$) is the index of singular values.

be observed, under this different heterogeneity scenario, FEDDECORR also yields noticeable and consistent improvements.

|  | $C = 2$ | $C = 3$ |
|---|---|---|
| FedAvg | 45.61 | 67.53 |
| + FEDDECORR | 47.63 | 74.51 |

Table 10: FEDDECORR yields noticeable and consistent improvements under another type of data heterogeneity.

## D    ADDITIONAL VISUALIZATIONS ON GLOBAL MODELS

In this section, we provide additional visualizations on global models with different federated learning methods, model architectures, and datasets. Through our extensive experimental results, we demonstrate that dimensional collapse is a general problem under heterogeneous data in federated learning.

### D.1    VISUALIZATION ON GLOBAL MODELS OF OTHER FEDERATED LEARNING METHODS

In the main text, we have shown that global models produced by FedAvg (McMahan et al., 2017) suffer stronger dimensional collapse with increasing data heterogeneity. To further show such dimensional collapse phenomenon is a general problem in federated learning, we visualized global models produced by other federated learning methods such as FedAvg with server momentum (Hsu et al., 2019), FedProx (Li et al., 2020), and MOON (Li et al., 2021b). Specifically, we follow the same procedure as in the main text and plot the singular values of covariance matrices of representations. Results are shown in Fig. 7. From the figure, one can see that all these three other methods also demonstrated the similar hazard of dimensional collapse as in FedAvg.

### D.2    VISUALIZATION ON GLOBAL MODELS OF OTHER MODEL ARCHITECTURES

In the main text, we have shown the dimensional collapse on global models caused by data heterogeneity with MobileNetV2. In this section, we perform the similar visualization based on other

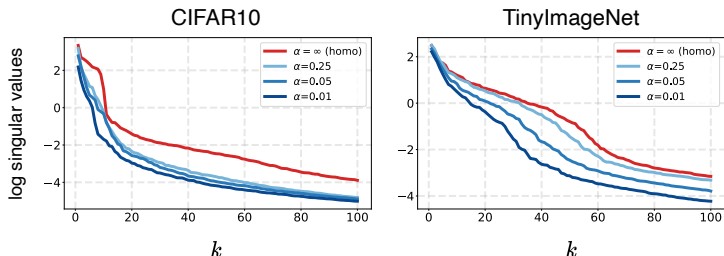

Figure 9: Data heterogeneity causes similar dimensional collapse on other datasets during federated learning. The x-axis ($k$) is the index of singular values.

model architectures such as ResNet32 and ResNet18. Note that ResNet32 is a narrower ResNet whose representation dimension is 64 and ResNet18 is a wider ResNet whose representation dimension is 64. We visualize the top 50 singular values for ResNet32 and the top 100 singular values for ResNet18. Results are shown in Fig. 8. From the figure, one can observe that heterogeneous data also lead to dimensional collapse on ResNet32 and ResNet18.

### D.3 VISUALIZATION ON GLOBAL MODELS OF OTHER DATASETS

In the main text, we use the CIFAR100 dataset for our visualizations. In this section, we perform similar visualizations with other datasets such as CIFAR10 and TinyImageNet. Results are shown in Fig. 9. From the figure, one can also observe that dimensional collapse results from data heterogeneity.

## E VISUALIZATION ON OTHER LOCAL CLIENTS

In the main text Fig. 2(b), under the four different degrees of data heterogeneity (i.e., $\alpha \in \{0.01, 0.05, 0.25, \infty\}$), we compare representations of local models of client 1 and empirically show how data heterogeneity affects representations produced by the local models. In this section, to further corroborate our conclusion, we follow the same procedure and visualize singular values of the covariance matrix of representations produced by local models trained on the rest of the 9 clients under the same $\alpha$'s. Results are shown in Fig. 10. From the results, we can obtain the similar observations as in Fig. 2(b) of the main text, namely that stronger data heterogeneity causes more severe dimensional collapse for local models.

## F HYPERPARAMETERS OF OTHER FEDERATED LEARNING METHODS

The regularization coefficient of FedProx (Li et al., 2020) $\mu$ is tuned across $\{10^{-4}, 10^{-3}, 10^{-2}, 10^{-1}\}$ and is selected to be $\mu = 10^{-3}$; the regularization coefficient of MOON (Li et al., 2021b) $\mu$ is tuned across $\{0.1, 1.0, 5.0, 10.0\}$ and is selected to be $\mu = 1.0$; the server momentum of FedAvgM (Hsu et al., 2019) $\rho$ is tuned across $\{0.1, 0.5, 0.9\}$ and is selected to be $\rho = 0.5$.

## G PSEUDO-CODE OF FEDDECORR

Here, we provide a pytorch-style pseudo-code for FEDDECORR in Alg. 1. All FEDDECORR-specific components are highlight in blue. As indicated in the pseudocode, the only additional operation of FEDDECORR is in adding a regularization term $L_{\text{FedDecorr}}(w, X)$ defined in Eqn. (8). This shows that FEDDECORR is an extremely convenient plug-and-play federated learning method.

## H STABILITY OF FEDDECORR REGULARIZATION LOSS

In this section, we first split CIFAR10 into 10 clients with $\alpha = 0.5$. Then, we plot how FedDecorr loss evolve within 10 local epochs for all the 10 clients in Fig. 11. All training configurations are the same as in the main paper. From the results, one can observe that the optimization process of FedDecorr loss is stable.

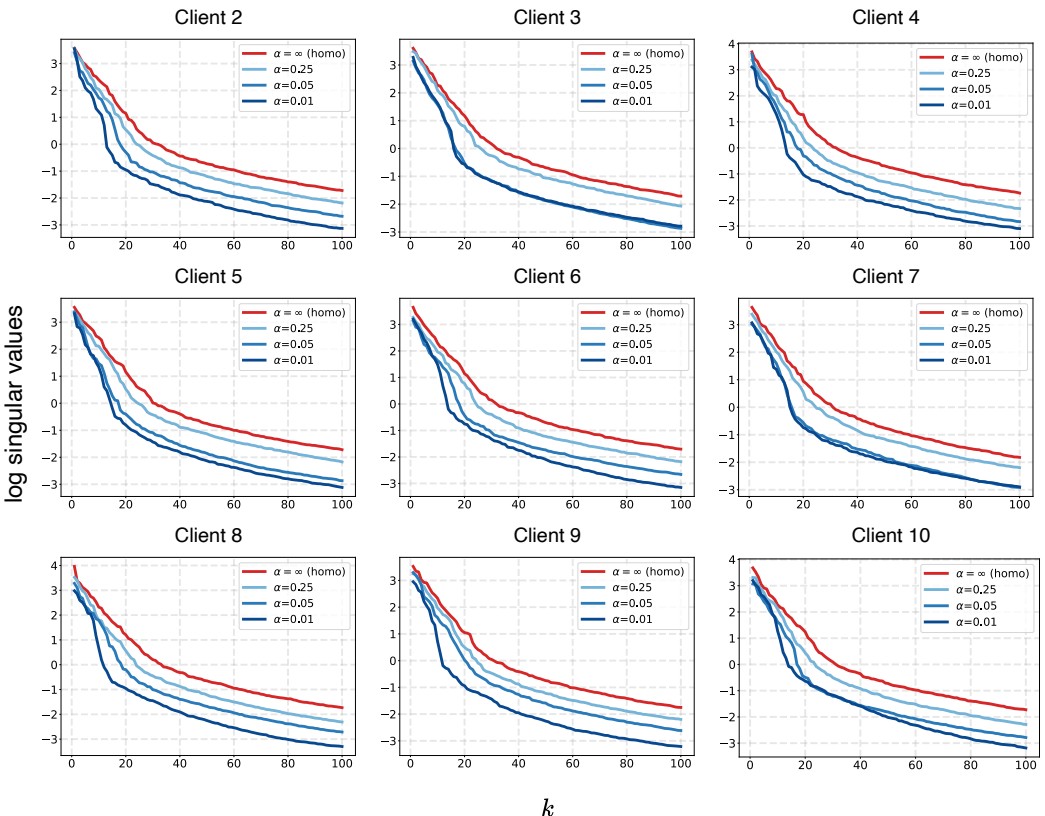

Figure 10: **Heterogeneous local training data cause dimensional collapse.** For each of the clients, given the four models trained under different degrees of heterogeneity, we plot the singular values of covariance matrix of representations in descending orders (the results of client 1 are shown in main text Fig. 2(b)). Representations are computed over the CIFAR100 test set. The $x$-axis ($k$) is the index of singular values and the $y$-axis is the logarithm of the singular values.

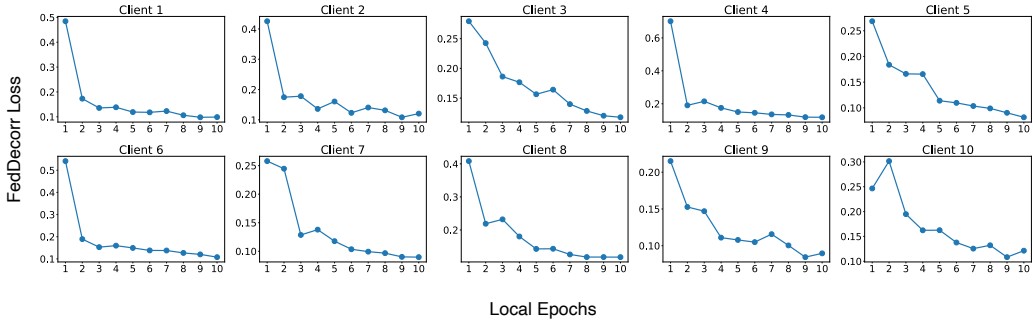

Figure 11: **How FedDecorr loss evolve within** 10 **local epochs.**

**Algorithm 1** PyTorch-style Pseudocode for FEDDECORR. (Blue highlights FEDDECORR-specific code)

```
def FedDecorrLoss(z):
    # N: batch size
    # d: representation dimension
    # z: a batch of representation, with shape (N, d)
    N,d = z.shape

    # z-score normalization
    z = (z - z.mean(0)) / z.std(0)

    # estimate correlation matrix
    corr_mat = 1/N*torch.matmul(z.t(), z)

    # calculate FedDecorr loss
    loss_fed_decorr = (corr_mat.pow(2)).mean()
    return loss_fed_decorr

def LocalTraining(train_loader, local_epochs, beta):
    for e in range(local_epochs):
        for data, targets in train_loader:
            # forward propagation.
            # given the batch of data, compute batch representations z and loss
            loss, z = ...

            loss += beta*FedDecorr(z)

            # back propagation and update local model parameters
            ...

def GlobalAggregation():
    # receiving models from each clients
    ...
    # aggregating local models with certain schemes
    ...
    # sending aggregated models back to clients
    ...

def main():
    # n_comm_round: number of communication rounds.
    # train_loader: data loader of training data.
    # n_local_epochs: number of local trainig epochs on each client.
    # beta: coefficient of the FedDecorr regularization.
    for comm in range(n_comm_round):
        LocalTraining(train_loader, n_local_epochs, beta)
        GlobalAggregation()
```

