# OpenReview forum: "Towards Understanding and Mitigating Dimensional Collapse in Heterogeneous Federated Learning"
_ICLR.cc/2023/Conference — ICLR 2023 poster_

### Official Review · Reviewer_h8K7 · 2022-10-20

**Confidence:** 4
**Correctness:** 3
**Technical Novelty And Significance:** 3
**Empirical Novelty And Significance:** 3
**Recommendation:** 6

**Clarity, Quality, Novelty And Reproducibility:**

Overall, this is an interesting experimental paper that uses simple observation of dimension collapse to propose new algorithms for mitigating data heterogeneity in FL. This paper is well-written with many supporting baselines and 3 vision datasets. It is well written but the code is not shared. Could the authors share the code in the supplementary, if possible? In this way people can confirm the reproducibility.

The idea is novel compared to existing frameworks, but it still needs more theoretical understanding.

**Strength And Weaknesses:**

Strengths:
1) Data heterogeneity is common in FL and this paper studies this problem from the perspective of singular values of features. This perspective is interesting and has not been studied before. The paper is well-motivated.
2) The proposed method is easy to understand, by equalizing the variance of singular values of the covariance matrix, with the Frobenius norm of the correlation matrix, the authors propose to use the F-norm as a regularization method. This is easy to implement and can be adapted to most existing FL methods, such as FedAvg, FedProx, MOON, etc.
3) This method has been tested on a variety of vision datasets, with different levels of data heterogeneity.

Weaknesses:
Although this paper proposes an interesting idea for mitigating the label shift problem, I still feel it needs some improvement.

1) Sec 3.3 seems unnecessary and is detached from the main paper. The linear network assumption seems too restrictive and does not provide us with new insights. One can simply obtain eq. (6) from the observation in Figure 2, and there is no need to use gradient flow dynamics.
2) Only Dirichlet partition has been considered. More testing needs to be done on other splits. Also, realistic partitions should be tested, such as FEMNIST, Shakespeare and StackOverflow.
3) The current method seem more like heuristics and it needs more understanding. For example, why is $\beta = 0.1$ more desirable? Why increasing $\beta$ can sometimes decrease the performance?
4) About baseline comparison, the authors should add more competitive baselines such as FedAdam [1] and FedLC [2] that aims to address the same data heterogeneity problem.


[1] Reddi, Sashank J., et al. "Adaptive Federated Optimization." International Conference on Learning Representations. 2020.
[2] Zhang, Jie, et al. "Federated Learning with Label Distribution Skew via Logits Calibration." International Conference on Machine Learning. PMLR, 2022.


**Summary Of The Paper:**

This paper studies dimension collapse in federated learning with label distribution shift, and methods on more to mitigate it. With the simple observation that the singular values of features are decaying fast for heterogeneous cases, the authors propose to minimize the corresponding variance, or equivalently, the Frobenius norm of the correlation matrix. This F-norm regularization has been tested on vision datasets with several baseline comparisons.

--post rebuttal--

Thanks to the authors for showing extra experiments on new data splits and datasets as well as comparison with new baselines. From the discussion, this paper is primarily experimental, and there is not much theoretical analysis on the explanation of $\beta$, or which algorithm  is the best when combined with DeCorr. These are some future directions.

**Summary Of The Review:**

In summary I would recommend weak accept as this paper identifies the problem of dimension collapse in FL and proposed a simple yet effective method to solve it. There are still some remaining issues with this method, such as baseline comparison, and deeper understanding beyond linear dynamics. The authors may start by proving faster convergence rate or better generalization error for FedDecorr, which could further improve the theoretical understanding. Also, it remains unclear to me why for TinyImageNet with $\alpha = \infty$ FedDecorr still has moderate improvement. Moreover, without FedDecorr, all the baseline algorithms do not beat FedAvg on CIFAR-100 and TinyImageNet by a large margin (1%), which raises questions for the correctness of the implementation.

---

> ### Author Response · Authors · 2022-11-11
> **Responses to Reviewer h8K7 (2/2)**
>
> **Q4: Comparison with more competitive baselines such as FedAdam and FedLC are needed.**
>
> **A4:** We first provide a comparison with FedLC [1]. Specifically, we apply FedLC on top of the FedAvg and compare the performance with FedDecorr. We use the CIFAR10 dataset with $\alpha \in \\{0.05, 0.1\\}$. All training configurations are the same as in the main paper for fair comparisons. Results are given in the following table, which shows that applying FedDecorr on the baseline yields more significant improvements compared to FedLC.
> ||$\alpha=0.05$|$\alpha=0.1$|
> |----|----|----|
> |FedAvg+FedLC|  67.96  |  77.86  |
> |**FedAvg+FedDecorr**|  **73.06**  |  **80.60**  |
>
>
> Next, we provide a comparison with FedAdam [2]. In this experiment, we apply FedDecorr on FedAdam. Similarly, we use CIFAR10 with $\alpha \in \\{0.05, 0.1\\}$. We found that FedAdam cannot converge (NaN loss) when setting the local epochs to be $10$ as in our paper. Therefore, following the experimental details specified in [2], we set the number of local epochs to be $1$ and use a ResNet18 with group normalization. We run FedAdam for $400$ communication rounds. The results are shown in the following table, which shows that FedDecorr improves over FedAdam, and the improvement is more pronounced for higher heterogeneity (smaller $\alpha$).
> ||$\alpha=0.05$|$\alpha=0.1$|
> |----|----|----|
> |FedAdam|  78.74  |  84.23  |
> |**FedAdam+FedDecorr**|  **81.63**  |  **85.13**  |
>
> **Q5: Why FedDecorr still has improvements for TinyImageNet with $\alpha=\infty$?**
>
> **A5:** As discussed in the second paragraph of Sec. 5.2 of the main paper, we conjecture that this is because TinyImageNet is much more complicated compared to the CIFAR datasets, and some other factors besides label heterogeneity may cause undesirable dimensional collapse in federated learning.
>
>
> **Q6: Why other baseline algorithms do not beat FedAvg on CIFAR-100 and TinyImageNet by a large margin ($1\\%$)?**
>
> **A6:** According to results of some other previous works (e.g., Table 2 of [3], Table 3 of [4]), the performances of these baseline algorithms in many cases also do not surpass that of the FedAvg baseline by a large margin. We are confident that our implementation is correct and as stated in the reproducibility statement, code for all experiments will be released.
>
>
>
> [1] Zhang, Jie, et al. Federated Learning with Label Distribution Skew via Logits Calibration. *International Conference on Machine Learning*. PMLR, 2022.
>
> [2] Sashank J. Reddi, et al. Adaptive Federated Optimization. In *International Conference on Learning Representations*, 2021.
>
> [3] Luo, Mi, et al. No fear of heterogeneity: Classifier calibration for federated learning with non-iid data. *Advances in Neural Information Processing Systems 34 (2021): 5972-5984*.
>
> [4] Qinbin Li, et al. Federated Learning on Non-IID Data Silos: An Experimental Study. *2022 IEEE 38th International Conference on Data Engineering (ICDE)*.
>
> [5] H. Brendan McMahan, et al. Communication-Efficient Learning of Deep Networks from Decentralized Data. In *Artificial intelligence and statistics*, pages 1273–1282. PMLR, 2017.
>
> [6] Hongyi Wang, et al. Federated learning with matched averaging. *International Conference on Learning Representations*, 2019.

---

> ### Author Response · Authors · 2022-11-11
> **Responses to Reviewer h8K7 (1/2)**
>
> Thanks for appreciating the contributions of our paper and for the constructive suggestions! Please see our responses to your questions below:
>
> **Q1: Sec 3.3 seems detached from the main paper.**
>
> **A1:** The FedDecorr method itself is indeed not directly related to the gradient flow dynamics. The purpose of Sec. 3.3 is to develop an intuitive theoretical understanding of the dimensional collapse phenomenon observed empirically in Sec. 3.2. Our analysis provides new theoretical insights as to how data heterogeneity exacerbates the undesirable dimensional collapse phenomenon. Although we only analyze linear neural networks, we believe that our analysis can serve as a concrete step towards theoretical understanding of general (nonlinear) neural networks.
>
> **Q2: More evaluations besides Dirichlet partition need to be done.**
>
> **A2:** In Sec. C.5 of Appendix, we have provided more results based on a different partition scheme introduced in [5]. Specifically, we split the CIFAR10 dataset across different clients such that each client only has a fixed number of classes $C$ (e.g., $C=2$ indicates each client only has data of two classes). In the Appendix, we have done experiments with $C \in \\{2,3\\}$. Here, we restate the results provided in the Appendix and further augment additional results with $C \in \\{4,5\\}$ in the following table. According to our results, improvements from FedDecorr is consistent under this different data partition scheme.
> ||$C=2$|$C=3$|$C=4$|$C=5$|
> |---|---|---|---|---|
> |FedAvg|45.61|67.53|74.93|84.69|
> |**+FedDecorr**|**47.63**|**74.51**|**78.45**|**87.49**|
>
>
> In addition, here we provide experimental results on the realistic partition of the Shakespeare dataset. Given the FedAvg baseline, we add the FedDecorr regularization on the output representations of the language model during local training. We use the heterogeneous split of Shakespeare, where each role is viewed as a local client. Following [6], we preprocess the Shakespeare dataset by filtering out the clients with less than $10,000$ datapoints and sampling a random subset of $66$ clients. We split $80\\%$ of the data of each client for training and merging the remaining data into a global test set. We experiment on a $1$-layer LSTM and a $2$-layer LSTM for this task. Results are given in the following table. All results are averaged over $3$ runs and are reported in mean$\pm$std. Our results show that FedDecorr can achieve improvements in the realistic partitions of Shakespeare.
> ||1-layer LSTM|2-layer LSTM|
> |---|---|---|
> |FedAvg|49.51 $\pm$ 0.04|49.75 $\pm$ 0.07|
> |**+FedDecorr**|**50.11 $\pm$ 0.17**|**50.56 $\pm$ 0.21**|
>
>
> **Q3: Why does increasing $\beta$ sometimes decrease the performance?**
>
> **A3:** Although decorrelation prevents dimensional collapse and improves heterogeneous federated learning, setting $\beta$ to be **too large** will slightly cause the degree of separation among representations of different classes to be deteriorated. This is because our decorrelation loss tends to slightly encourage representations of all data to be spread out in the space instead of being clustered to different class centroids. Therefore, it is recommanded NOT to set the $\beta$ to be too large so that we can prevent excessive dimensional collapse while enjoy good class separation in terms of the representations.

---

### Official Review · Reviewer_QKrC · 2022-10-23

**Confidence:** 3
**Correctness:** 4
**Technical Novelty And Significance:** 3
**Empirical Novelty And Significance:** 3
**Recommendation:** 8

**Clarity, Quality, Novelty And Reproducibility:**

The paper is clear in its arguments and is of good quality. The approach to the best of my knowledge is novel. I have several questions, however, regarding the experimental results.

1. In table 4 on Page 9, the results for different client sampling rates are shown. Can the authors explain why performance improves when the local epoch is increased or cite other sources as counter examples? As per my understanding, increasing local epochs increases client drift and should degrade performance.[1]
2. Why does correlation matrix work better than covariance matrix?
3. How stable is the added regression term?
4. Can the proposed algorithm work for other tasks like natural language processing, eg masked language modeling?


[1] Wang, Jianyu, et al. "A field guide to federated optimization." arXiv preprint arXiv:2107.06917 (2021).

**Strength And Weaknesses:**

The paper identifies a key issue in data heterogeneous federated learning. The overview of dimensional collapse in FL is a significant stepping stone for future research. The proposed algorithm is simple yet effective. The paper also shows theoretical and experimental results which makes their arguments strong. I do have some suggestions.

1. I believe it would be a good to see how the algorithm behaves when we vary the client sampling rate in addition to the number of clients.
2. Add a convergence analysis for the algorithm if tractable.
3. An ablation study studying how the added regression evolves over time is important.
4. Add more tasks besides image classification.

**Summary Of The Paper:**

The paper identifies a key bottleneck in highly heterogeneous federated systems, namely dimensional collapse. The authors propose a simple but novel technique for mitigating dimensional collapse by adding a regularization term during local training. The paper shows both theoretical and empirical analysis to support their results.

**Summary Of The Review:**

The paper points out a key problem in highly heterogeneous federated systems and proposes a novel approach to solving it. The study is a good direction for future research as well. The authors back up their arguments with both rigorous theoretical and experimental results on image classification tasks. I do have reservations regarding the stability of the algorithm and its effectiveness in other domains. Nevertheless, the results are more or less rigorous and show it is an effective technique.

---

> ### Author Response · Authors · 2022-11-11
> **Responses to Reviewer QKrC (2/2)**
>
> **Q5: Why performance improves when the local epoch is increased?**
>
> **A5:**  According to empirical results in previous literature [1,2], local epochs being too large indeed causes more severe client drift and hence degrades the performance. However, if the number of local epochs is too small, then the model cannot converge well given a fixed number of communication rounds. As a result, there is normally a “sweet spot” in terms of the number of local epochs, which strikes a balance between convergence of the optimization procedure and client drift. According to our results, setting the number of local epochs to be $E=10$ consistently produces (near) optimal results. This analysis is also given in Section 5.5 of our main paper. Similar empirical observations have also been obtained in previous works (e.g., Figure 7 (a-c) of [1]).
>
>
> **Q6: Why does correlation matrix work better than covariance matrix?**
>
> **A6:** As discussed in Appendix C.2, we posit that this is because the correlation matrix is a normalized version of covariance matrix (i.e., computing correlation matrix would need to first apply the usual Z-score normalization on embeddings). Using such a normalization, the optimization procedure can be more stable and hence, leads to the desired representations. If this normalization is not done, some features with large numerical values might be significantly over-emphasized relative to those with small numerical values.
>
>
> **Q7: How stable is the added regularization term?**
>
> **A7:** As described in our answer to Q3, the FedDecorr loss demonstrates relatively stable behavior across the optimization process.
>
> In addition, in Table 1 and Table 2 of the main paper, to assess the stability of our numerical results, the standard deviations of accuracies over 3 runs are shown. According to the results, adding FedDecorr will not significantly increase the standard deviation of the results. In fact, in many cases (e.g., CIFAR10 w/ beta=0.05 and FedAvg baseline), the standard deviations of the accuracies even decrease significantly.
>
> Furthermore, in Figure 4 of the main paper, we plot the evolution of the global model accuracy throughout federated learning process. As illustrated in this figure, adding FedDecorr leads to robust and stable convergence.
>
> Finally, in our experiments in Figure 5 of the main paper, we have shown that varying the coefficient of FedDecorr ($\beta$) within a certain range will not drastically affect the performance.
>
> All these results demonstrate that our FedDecorr is an algorithm that is stable in terms of statistical variations, optimization convergence, as well as different choices of the hyperparameter $\beta$.
>
>
> [1] Qinbin Li, et al. Model-Contrastive Federated Learning. In *Proceedings of the IEEE/CVF Conference on Computer Vision and Pattern Recognition*, pages 10713–10722, 2021.
>
> [2] Qinbin Li, et al. Federated Learning on Non-IID Data Silos: An Experimental Study. *2022 IEEE 38th International Conference on Data Engineering (ICDE)*.
>
> [3] Hongyi Wang, et al. Federated learning with matched averaging. *International Conference on Learning Representations*, 2019.

---

> ### Author Response · Authors · 2022-11-11
> **Responses to Reviewer QKrC (1/2)**
>
> Thanks for appreciating the contributions of our paper and for the constructive suggestions! Please see our responses to your questions below:
>
> **Q1: How FedDecorr would behave under varying client sampling rate?**
>
> **A1:** In the main paper, we have used a sampling rate of $20\\%$ when splitting TinyImageNet into 50 clients. Here, we provide additional results by using the sampling rate of $40\\%$ and $60\\%$. We experiment under $\alpha=0.1$. Test accuracies are shown in the following table. From the results, we can consistently observe improvements of at least $8\\%$ across all tested sampling rate. This shows that FedDecorr is robust under different sampling rates.
> ||sample rate=$20\\%$|$40\\%$|$60\\%$|
> |---|---|---|---|
> |FedAvg|28.88|29.51|30.6|
> |**+FedDecorr**|**36.67**|**38.57**|**40.09**|
>
>
>
> **Q2: Whether a convergence analysis is tractable?**
>
> **A2:** To the best of our knowledge, it might be intractable at present to analyze the convergence behavior for the decorrelation loss. However, we would like to highlight that according to our empirical observations from Figure 4 of the main paper, applying our FedDecorr can result in smooth and effective convergence.
>
>
>
> **Q3: How FedDecorr loss evolves over time?**
>
> **A3:** Here, we split CIFAR10 dataset into $10$ clients with $\alpha=0.5$. Then, we run FedDecorr on each client for $10$ local epochs. All training configurations are the same as in the main paper. We show how FedDecorr loss evolves for the local model of the first client in the following table. Results of other clients are similar and are plotted and updated in Section H of the Appendix (see Figure 11 of the Appendix). From the results, one can observe that the optimization process of FedDecorr loss is rather stable.
> |Local Epoch|1|2|3|4|5|6|7|8|9|10|
> |---|---|---|---|---|---|---|---|---|---|---|
> |FedDecorr loss|0.4845|0.1728|0.1354|0.1381|0.1188|0.1178|0.1227|0.1060|0.0979|0.0986|
>
>
>
>
> **Q4: Add more tasks besides image classification.**
>
> **A4:** Here, we conduct experiments on a language modeling dataset: Shakespeare. Given the FedAvg baseline, we add the FedDecorr regularization on the output representations of the language model during local training. We use the heterogeneous split of Shakespeare, where each role is viewed as a local client. Following [3], we preprocess the Shakespeare dataset by filtering out the clients with less than $10,000$ datapoints and sampling a random subset of $66$ clients. We split $80\\%$ of the data of each client for training and merging the remaining data into a global test set. We experiment on a $1$-layer LSTM and a $2$-layer LSTM for this task. Test accuracies are given in the following table. All results are averaged over $3$ runs and are reported in mean$\pm$std. Our results show that FedDecorr can also achieve consistent improvements in natural language tasks (in addition to vision tasks).
>
> ||1-layer LSTM|2-layer LSTM|
> |---|---|---|
> |FedAvg|49.51 $\pm$ 0.04|49.75 $\pm$ 0.07|
> |**+FedDecorr**|**50.11 $\pm$ 0.17**|**50.56 $\pm$ 0.21**|

---

### Official Review · Reviewer_mWEv · 2022-10-26

**Confidence:** 3
**Correctness:** 4
**Technical Novelty And Significance:** 4
**Empirical Novelty And Significance:** 3
**Recommendation:** 8

**Clarity, Quality, Novelty And Reproducibility:**

The observation and the proposed algorithm are novel and clear. However, the source code is not provided.


**Strength And Weaknesses:**

Strengths

- This paper is easy to follow.

- The proposed algorithm is well-motivated and performed very well.

- The dimensional collapse is an interesting observation.

Weakness

- More recent FL algorithms should be discussed and compared.

- More experiments with other data sets and neural net architectures might be required.

**Summary Of The Paper:**

This paper considers the federated learning problem with data heterogeneity. From experiments, the authors find an interesting phenomenon, dimensional collapse. The global model suffers from dimensional collapse in the feature space such that the covariance matrix of the representations of the global model becomes approximately low-rank. The model collapse also happens in local models. The authors claim that the model collapse is one of the main reasons that degrade the FL performance. To remedy this problem, they propose FedDecorr, which applies a Frobenius norm regularization term during local training, making the distributions of singular values more uniform. FedDecorr outperforms baselines on standard benchmark datasets.


**Summary Of The Review:**

This paper proposes a simple yet efficient algorithm for FL with a good justification.

---

> ### Author Response · Authors · 2022-11-11
> **Responses to Reviewer mWEv**
>
> Thanks for appreciating the contributions of our paper and for the constructive suggestions! Please see our responses to your questions below:
>
>
> **Q1: More recent FL algorithms should be discussed and compared.**
>
> **A1:** In Table 8 and Table 9 of the Appendix, we have provided additional empirical comparisons with two more baselines---Scaffold and FedNova. As shown in these tables, across various datasets and degrees of heterogeneity, the highest accuracies are achieved when adding FedDecorr on top of a certain baseline FL method.
>
> To augment the results, here we compare FedDecorr with another recent federated learning method: FedLC [1]. Specifically, we apply FedLC on top of the FedAvg and compare the performance with FedDecorr. We use CIFAR10 with $\alpha \in \\{0.05, 0.1\\}$. The temperature of FedLC is tuned across $\tau \in \\{0.001, 0.01, 0.1, 1.0\\}$. All training configurations are the same as in the main paper to ensure that the comparisons are fair. Results are given in the following table, which clearly show that applying FedDecorr on top of the baseline leads to more significant improvements.
>
> ||$\alpha=0.05$|$\alpha=0.1$|
> |----|----|----|
> |FedAvg+FedLC|  67.96  |  77.86  |
> |**FedAvg+FedDecorr**|  **73.06**  |  **80.60**  |
>
>
> **Q2: More experiments with other network architectures might be required.**
>
> **A2:** As shown in Table 7 of the Appendix, we have run the same experiments on two additional neural network architecture (Resnet32 and Resnet18) with the CIFAR10 dataset with heterogeneity parameter $\alpha=0.05$. Here we restate our results (that were stated in the Appendix) and further augment the results with $\alpha=0.1$ on the same dataset in the following table. According to our results, applying FedDecorr **consistently** improves over the baseline under **different network architectures**.
>
> ||MobileNetV2|ResNet18|ResNet32|
> |-----|-----|-----|-----|
> |FedAvg ($\alpha=0.1$)|76.28|82.32|73.22|
> |**+FedDecorr ($\alpha=0.1$)**|**80.60**|**83.59**|**74.75**|
> |FedAvg ($\alpha=0.05$)|64.85| 71.51|65.76|
> |**+FedDecorr ($\alpha=0.05$)**|**73.06**|**76.54**|**67.21**|
>
> **Q3: Regarding the source code availability.**
>
> **A3:** As stated in our reproducibility statement, source code for all experiments will be released.
>
> [1] Zhang, Jie, et al. Federated Learning with Label Distribution Skew via Logits Calibration. *International Conference on Machine Learning*. PMLR, 2022.

---

### Official Review · Reviewer_deuX · 2022-11-04

**Confidence:** 4
**Correctness:** 3
**Technical Novelty And Significance:** 2
**Empirical Novelty And Significance:** 2
**Recommendation:** 5

**Clarity, Quality, Novelty And Reproducibility:**

The paper is overall well-written and enough details about experiments in main body and appendix are included for reproducibility purpose.

**Strength And Weaknesses:**

The paper makes an interesting observation and has some merits, but

On the empirical side,

1. There is a discrepancy between experiments the observation made from and proposed algorithmic solution: the way the heterogeneity is introduced among clients to infer empirical observation include discrepancy among both covariates and labels, while the proposed algorithm only tackles representations by regularizing  covariance matrix of the representations. This left me wondering about the contribution of each source of heterogeneity on dimensional collapse.

2. Also, in the results reported in Table 4, it seems as the number of local updates increases, the proposed algorithm does not show significant improvement over baseline (FedAvg). Since in federated settings the number of local updates could be much larger than 20, it seems the proposed regularization becomes less beneficial (it is conjectured by authors that the dimensional collapse of the global model stems from local models).

3. Finally, it would be nice to see how the observed  phenomena manifest itself in other algorithms where a drift/diversity mitigation schema is employed. This could potentially better illustrate the role of heterogeneity in dimensional collapse and the effectiveness of the proposed method.

On the theoretical side, the analysis is limited to linear neural networks, does not take into account the local updating and aggregation schema, and more importantly does not clearly illustrate how covariance regularization can improve accuracy as it analyzes the gradient flow of optimization which does not necessarily translate to a good generalization in the presence of data heterogeneity.


**Summary Of The Paper:**

This paper makes an interesting observation in learning from heterogeneous data sources by empirically illustrating that representations tend to reside in a lower-dimensional space.  To overcome this issue, authors propose FEDDECORR algorithm that encourages the Frobenius norm of the correlation matrix of representations at each client (penultimate layer of a neural network) to be small.


**Summary Of The Review:**

The paper makes an interesting observation and algorithmic solution, but the theoretical results and empirical results need to be strengthened to resolve  issues discussed above.

---

> ### Author Response · Authors · 2022-11-11
> **Response to Reviewer deuX**
>
> Thanks for appreciating the contributions of our paper and for the constructive suggestions! Please see our responses to your questions below:
>
> **Q1: Contribution of each source of heterogeneity on dimensional collapse?**
>
> **A1:** We would like to kindly point out that in line with many previous works [1-5], we only focus on **label heterogeneity** in federated learning in this paper. All our empirical observations, theoretical analyses, and experiments are centered around label heterogeneity. Specifically, in our paper, we first observe and analyze how label heterogeneity cause undesired dimensional collapse, which motivates the design of our FedDecorr method. Therefore, our initial empirical observations dovetail nicely with our proposed method. Whether or not covariate/domain shift would also result in the undesired dimensional collapse phenomenon is indeed an interesting topic, and we leave it for future exploration.
>
> **Q2: Improvements under larger local epochs are smaller.**
>
> **A2:** We would first like to emphasize that although larger local epochs sometimes leads to slight decreases in the improvements produced by FedDecorr, the improvements are still significant in most cases. For example, when setting the number of local epochs $E=20$, 3 out of the 4 ablated cases observe accuracy improvements over $2\%$.
>
> Secondly, we provide additional numerical results with local epochs $E=30$. Results are given in the following table, which show that FedDecorr still yields significant improvements when we set $E>20$.
> ||CIFAR100|CIFAR100|TinyImageNet|TinyImageNet|
> |---|---|---|---|---|
> ||$\alpha=0.05$|$\alpha=0.1$|$\alpha=0.05$|$\alpha=0.1$|
> |FedAvg|56.45|64.29|28.64|37.41|
> |+FedDecorr|59.24|65.86|34.12|40.42|
>
>
> Finally, one common observation in Federated Learning is that local epochs being too large can unfortunately result in severe local client drift, which adversely affects the performance of the global model. This is shown in both our experiments (e.g., Table 4 of our main paper) and previous works (e.g., Figure 7 (a-c) of [5] and Figure 9 (a) of [6]), where setting local epochs to be too large (e.g., larger than 20) degrades accuracy. Usually, the (near) optimal local epochs observed for the datasets we experimented on is $E=10$.
>
>
> **Q3: Whether dimensional collapse are observed in FL methods with heterogeneity mitigation schemes?**
>
> **A3:** In Appendix D (Figure 7 of our Appendix), we have already provided additional visualizations for other FL methods with heterogeneity mitigation schemes (e.g., FedProx, MOON). According to our visualizations, dimensional collapse from data heterogeneity can still be clearly observed even for FedProx and MOON. This clearly demonstrates that the undesired dimensional collapse problem in FL has indeed been overlooked by previous methods, which provides even stronger motivation for the use of our proposed method FedDecorr.
>
>
> **Q4: Limitations on theoretic analysis.**
>
> **A4:** We admit that our theoretical analysis does not provide a definitive connection between representation decorrelation and final accuracy improvements as current analytical techniques are restricted to linear neural networks. However, we would like to emphasize that the key contribution of our work is the observation of dimensional collapse in heterogeneous federated learning and the proposal of the FedDecorr regularizer to mitigate this adverse phenomenon. The purpose of our theoretical analysis is simply to provide an auxiliary understanding of why label heterogeneity causes more severe dimensional collapse. More rigorous theoretical understanding (e.g., for general nonlinear neural networks) is left for future works.
>
>
> [1] H. Brendan McMahan, et al. Communication-Efficient Learning of Deep Networks from Decentralized Data. In *Artificial intelligence and statistics*, pages 1273–1282. PMLR, 2017.
>
> [2] Tian Li, et al. Federated optimization in heterogeneous networks. In *Proceedings of Machine Learning and Systems*, 2:429–450, 2020.
>
> [3] Sai Praneeth Karimireddy, et al. SCAFFOLD: Stochastic Controlled Averaging for Federated Learning. In *International Conference on Machine Learning*, pages 5132–5143. PMLR, 2020.
>
> [4] Sashank J. Reddi, et al. Adaptive Federated Optimization. In *International Conference on Learning Representations*, 2021.
>
> [5] Qinbin Li, et al. Model-Contrastive Federated Learning. In *Proceedings of the IEEE/CVF Conference on Computer Vision and Pattern Recognition*, pages 10713–10722, 2021.
>
> [6] Qinbin Li, et al. Federated Learning on Non-IID Data Silos: An Experimental Study. *2022 IEEE 38th International Conference on Data Engineering (ICDE)*.

---

> ### Author Response · Authors · 2022-11-16
> **Thanks for your efforts in reviewing our paper!**
>
> Dear Reviewer deuX,
>
> Since the author-reviewer discussion deadline is approaching, we would like to follow up with you to see if you have any more suggestions.
> In our previous responses to you, we have answered your questions on
> 1) the basic setups of our observations and experiments;
> 2) improvements under larger local epochs;
> 3) whether dimensional collapse is also observed in other FL methods with heterogeneity mitigation schemes;
> 4) limitations on theoretic analysis;
>
> Thanks again for your time and efforts in reviewing our work.
>
> Best,
>
> Authors of paper 1787

---

### Decision · Program_Chairs · 2023-01-20

**Decision:**

Accept: poster

**Justification For Why Not Higher Score:**

We're mostly positive but theory or cross-modality would need to be improved for reaching oral-level contribution.

**Justification For Why Not Lower Score:**

No major concerns remained after feedback

**Metareview: Summary, Strengths And Weaknesses:**

The paper studies federated learning on the realistic case of heterogeneous data, where client drift happens. In this setting of drift, it identifies undesired representation collapse into low-dimensional subspaces, and proposes to alleviate this with an algorithm based on regularizing representations for small correlations. Reviewers have consensus that they appreciate the interesting observations and simple new algorithm, while some questions remained if it would generalized to non-image data modalities too.

We hope the authors will incorporate the several points mentioned by the reviewers in the final version.

As another small comment I would also suggest to discuss on the phenomenon not happening for very small numbers of local updates (Table 4), as if only a single local batch is used then there will be no drift at all. Currently it is not super clear from which level of drift the phenomenon starts appearing (it is shown that it does after a full local epoch though)

**Note From Pc:**

if the above contains the word "oral" or "spotlight" please see: "oral" presentation means -> notable-top-5% and "spotlight" means -> notable-top-25%. As stated in our emails, we are disassociating presentation type from AC recommendations